# DPA: A One-stop Metric to Measure Bias Amplification in Classification Datasets

**Bhanu Tokas**$^*$
Arizona State University
bhanu.tokas@asu.edu

**Rahul Nair**$^*$
Arizona State University
rnair21@asu.edu

**Hannah Kerner**
Arizona State University
hkerner@asu.edu

## Abstract

Most ML datasets today contain biases. When we train models on these datasets, they often not only learn these biases but can worsen them — a phenomenon known as bias amplification. Several co-occurrence-based metrics have been proposed to measure bias amplification in classification datasets. They measure bias amplification between a protected attribute (e.g., gender) and a task (e.g., cooking). These metrics also support fine-grained bias analysis by identifying the direction in which a model amplifies biases. However, co-occurrence-based metrics have limitations — some fail to measure bias amplification in balanced datasets, while others fail to measure negative bias amplification. To solve these issues, recent work proposed a predictability-based metric called leakage amplification (LA). However, LA cannot identify the direction in which a model amplifies biases. We propose Directional Predictability Amplification (DPA), a predictability-based metric that is (1) directional, (2) works with balanced and unbalanced datasets, and (3) correctly identifies positive and negative bias amplification. DPA eliminates the need to evaluate models on multiple metrics to verify these three aspects. DPA also improves over prior predictability-based metrics like LA: it is less sensitive to the choice of attacker function (a hyperparameter in predictability-based metrics), reports scores within a bounded range, and accounts for dataset bias by measuring relative changes in predictability. Our experiments on well-known datasets like COMPAS (a tabular dataset), COCO, and ImSitu (image datasets) show that DPA is the most reliable metric to measure bias amplification in classification problems. To compare DPA with existing bias amplification metrics, we released a one-stop library of major bias amplification metrics at https://github.com/kerner-lab/Bias-Amplification.

## 1 Introduction

Machine learning models should perform fairly across demographics, genders, and other groups. However, ensuring fairness is challenging when training datasets are biased, as is the case with many datasets. For instance, in the ImSitu dataset [1], $67\%$ of the images labeled "cooking" feature females, indicating a gender bias that women are more likely to be associated with cooking than men [2]. Given a biased training set, it is not surprising for a model to learn these dataset biases. Surprisingly, models not only learn dataset biases but can also amplify them [2, 3, 4, 5, 6]. In the example from ImSitu, where females and cooking co-occurred $67\%$ of the time, bias amplification occurs when $> 67\%$ of the images predicted as cooking feature females.

Several co-occurrence-based metrics ($BA_{\rightarrow}$, $Multi_{\rightarrow}$, $BA_{MALS}$) have been proposed to measure bias amplification in classification datasets. They measure the bias amplification between a protected attribute (e.g., gender), denoted as $A$, and a task (e.g., cooking), denoted as $T$ [2, 3, 4]. If $A$ and

---

$^*$These authors contributed equally to this work.

39th Conference on Neural Information Processing Systems (NeurIPS 2025).

$T$ co-occur more than random in the training dataset, these metrics measure how much more they co-occur in the model's predictions. For instance, if the co-occurrence between females ($A$) and cooking ($T$) is 67% in the training dataset and 90% at test time, the bias amplification value is 23%.

Co-occurrence-based metrics help with a fine-grained bias analysis as they can identify the direction in which a model amplified biases. In the cooking example from ImSitu, metrics like $BA_\rightarrow$ and $Multi_\rightarrow$ can measure how much a model amplified the bias towards predicting women as cooking ($A \rightarrow T$) and towards predicting cooks as women ($T \rightarrow A$). Such disentanglement of bias amplification is important, as it helps practitioners come up with targeted bias intervention efforts.

However, co-occurrence-based metrics have a limitation: they cannot measure bias amplification if $A$ is balanced with $T$. Metrics like $BA_\rightarrow$ and $BA_{MALS}$ assume that if attributes and tasks are balanced in the training dataset, there are no dataset biases to amplify. Previous works have shown that simply balancing attributes and tasks does not ensure an unbiased dataset. Biases may emerge from parts of the dataset that are not annotated [7].

Suppose we balance imSitu such that 50% of the images labeled "cooking" feature females. Now, assume that cooking objects in ImSitu, like hairnets, are not annotated. If most of the cooking images with females have hairnets, while most of the cooking images with males do not, the model may learn a spurious correlation between hairnets, cooking, and females. Hence, the model may more often predict the presence of a female when cooking images have hairnets in the test set, leading to bias amplification between females and cooking. However, since gender appears balanced with respect to the cooking labels, $BA_\rightarrow$ and $BA_{MALS}$ would report 0 bias amplification. While other co-occurrence metrics like $Multi_\rightarrow$ may report non-zero bias amplification for balanced datasets, these values are often misleading, as $Multi_\rightarrow$ cannot capture negative bias amplification scenarios.

To correctly measure bias amplification in balanced datasets (by accounting for biases from unanno­tated elements), Wang et al. [7] proposed a predictability-based metric called leakage. They defined dataset leakage ($\lambda^\mathrm{D}$), which refers to how well attribute $A$ can be predicted from true task labels $T$, and model leakage ($\lambda^\mathrm{M}$), which refers to how well $A$ can be predicted from a model's task predictions $\hat{T}$. They defined bias amplification as $\lambda^\mathrm{M} - \lambda^\mathrm{D}$. $\lambda^\mathrm{D}$ and $\lambda^\mathrm{M}$ are measured using a separate attacker function trained to predict $A$. In this work, we refer to Wang et al.'s method of calculating bias amplification as leakage amplification ($LA$). Since $LA$ focuses on predictability, it can measure bias amplification in balanced and unbalanced datasets. However, it cannot identify the direction in which a model amplifies biases ($A \rightarrow T$ and $T \rightarrow A$ disentanglement is not possible with $LA$).

We do not have one metric that can (1) correctly report positive and negative bias amplification scenarios, (2) accurately measure bias amplification for both balanced and unbalanced datasets, and (3) identify the direction in which a model amplifies biases. Practitioners need to evaluate their models on multiple bias amplification metrics to assess these properties, making the process tedious. We propose Directional Predictability Amplification ($DPA$), a one-stop predictability-based metric that addresses all these properties of bias amplification. In addition to being a one-stop metric, $DPA$ addresses several issues found in earlier predictability-based metrics like $LA$, including issues such as reporting unbounded bias amplification values, measuring absolute change in predictability instead of relative change, and being highly sensitive to attacker function.

Our key contributions: **(1)** $DPA$ is the only directional metric that can accurately identify positive and negative bias amplification in both balanced and unbalanced classification datasets. **(2)** $DPA$ reports bias amplification values within a bounded range of $[-1, 1]$, enabling easy comparison across different models **(3)** $DPA$ accounts for the original bias in the dataset as it measures relative change in predictability (instead of absolute change). **(4)** $DPA$ is minimally sensitive to the choice of attacker function. This eliminates the pain of choosing a suitable attacker to measure bias amplification.

## 2   Related Work

**Co-occurrence for Bias Amplification** *Men Also Like Shopping* ($BA_{MALS}$) [2] proposed the first metric for bias amplification. The proposed metric measured the co-occurrences between protected attributes $A$ and tasks $T$. For any $T - A$ pairs that showed a positive correlation (i.e., the pair occurred more frequently than independent events) in the training dataset, it measured how much the positive correlation increased in model predictions.

Wang and Russakovsky [3] generalized the $BA_{MALS}$ metric to also measure negative correlation (i.e., the pair occurred less frequently than independent events). Further, Wang and Russakovsky [3] changed how the positive bias is defined by comparing the independent and joint probability of a pair. But, both $BA_{MALS}$ [2] and $BA_\rightarrow$ [3] could only work for $T - A$ pairs when $T$, $A$ were singleton sets (e.g., {Basketball} & {Male}). Zhao et al. [4] extended the metric proposed by Wang and Russakovsky [3] to allow $T - A$ pairs when $T$, $A$ are non-singleton sets (e.g., {Basketball, Sneakers} & {African-American, Male}).

Lin et al. [8] proposed a new metric called bias disparity to measure bias amplification in recommender systems. Foulds et al. [9] measured bias amplification using the difference in "differential fairness", a measure of the difference in co-occurrences of $T - A$ pairs across different values of $A$. Seshadri et al. [10] measured bias amplification for text-to-image generation using the increase in percentage bias in generated vs. training samples.

**Bias Amplification in Balanced Datasets** Wang et al. [7] identified that $BA_{MALS}$ [2] failed to measure bias amplification for balanced datasets. They proposed a metric that we refer to as leakage amplification that could measure bias amplification in balanced datasets. While some of the previously discussed metrics [9, 10, 8, 4] can measure bias amplification in a balanced dataset, these metrics do not work for continuous variables, because they use co-occurrences to quantify biases.

Leakage amplification quantifies biases in terms of predictability, i.e., how easily a model can predict the protected attribute $A$ from a task $T$. Attacker functions ($f$) are trained to predict the attribute ($A$) from the ground-truth observations of the task ($T$) and model predictions of the task ($\hat{T}$). The relative performance of $f$ on $T$ vs. $\hat{T}$ represents the leakage of information from $A$ to $T$.

As the attacker function can be any kind of machine learning model, it can process continuous inputs, text, and images. This flexibility gives leakage amplification a distinct advantage over co-occurrence-based bias amplification metrics. Subsequent work used leakage amplification for quantifying bias amplification in image captioning [11].

**Capturing Directionality in Bias Amplification** While previous metrics, including leakage amplification [7], could detect the presence of bias, they could not explain its directionality. Wang and Russakovsky [3] was the first to introduce a directional bias amplification metric, $BA_\rightarrow$. However, the metric only works for unbalanced datasets. Zhao et al. [4] proposed a new metric, $Multi_\rightarrow$, to measure directional bias amplification for multiple attributes and balanced datasets. Still, the metric cannot distinguish between positive and negative bias amplification, as shown in section A. This lack of sign awareness makes $Multi_\rightarrow$ unsuitable for many use cases.

In summary, no existing metric can measure the positive and negative directional bias amplification in a balanced dataset, as shown in Table 1.

Table 1: We compare desirable properties of bias amplification metrics. Only $DPA$ has all three.

| Method | Balanced Datasets | Directional | Negative Amp. |
|---|---|---|---|
| $BA_{MALS}$ | ✗ | ✗ | ✓ |
| $BA_\rightarrow$ | ✗ | ✓ | ✓ |
| $Multi_\rightarrow$ | ✓ | ✓ | ✗ |
| $LA$ | ✓ | ✗ | ✓ |
| $DPA$ (Ours) | ✓ | ✓ | ✓ |

## 3 Directional Predictability Amplification

In this section, we explain how our predictability-based metric, $DPA$, measures directional bias amplification ($A \rightarrow T$ and $T \rightarrow A$). We also show how $DPA$ is more robust and easier to interpret compared to previous predictability-based metrics like $LA$. We demonstrate the other two properties of $DPA$: (1) its ability to quantify both positive and negative bias amplification, and (2) its efficacy across both balanced and unbalanced classification datasets, through our experiments (Section 4).

### 3.1 Problem Formulation and Proposed Method

Before introducing our proposed metric, we outline the problem setup. Consider a classification dataset consisting of images $I$, where each image is annotated with task labels $T$ and protected attributes $A$. A model $M$ processes the input images $I$ to predict tasks $\hat{T}$ and attributes $\hat{A}$.

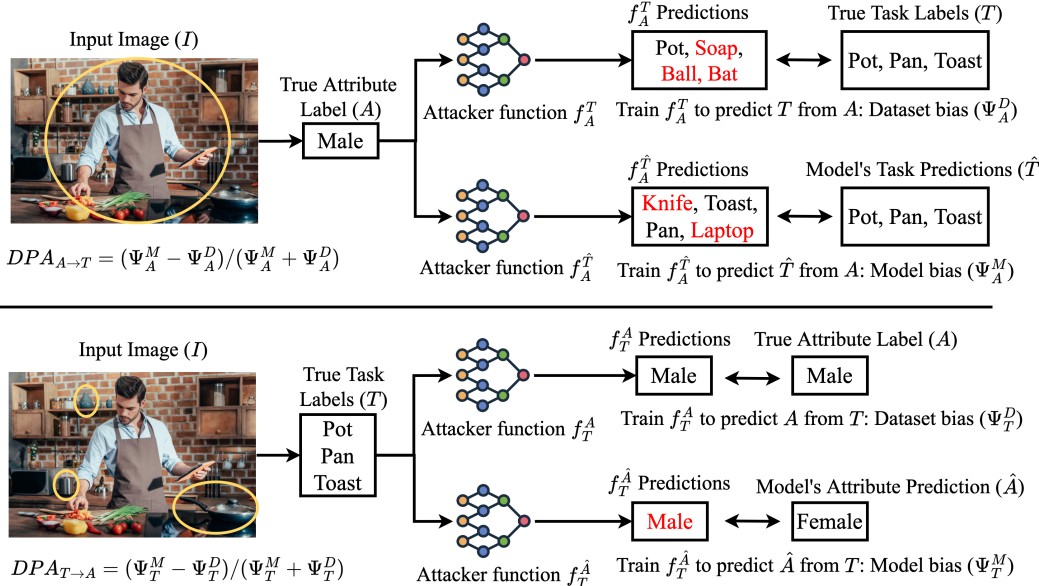

Figure 1: $A \rightarrow T$ and $T \rightarrow A$ bias amplification using our predictability-based metric $DPA$

To measure bias amplification, we quantified how much $A$ influences $T$ ($A \rightarrow T$ bias) and how much $T$ influences $A$ ($T \rightarrow A$ bias) in both the true labels (dataset bias) and the model $M$'s predictions (model bias). While previous directional metrics, such as $BA_\rightarrow$ [3] and $Multi_\rightarrow$ [4], used conditional probabilities to quantify dataset and model bias, we used the concept of predictability.

To measure dataset bias in $A \rightarrow T$ ($\Psi_A^D$), we trained an attacker function $f_A^T$ that predicts true task labels ($T$) using the true attribute label ($A$) as an input. $f_A^T$ is any arbitrary model, such as an SVM, a decision tree, or a neural network. We measured attacker function performance using a quality function $Q$. Accuracy and F-1 scores are examples of some popular quality functions. If $f_A^T$ predicts $T$ from input $A$ with high accuracy, it shows that the true attribute label ($A$) has high predictability for true task labels ($T$), indicating a high dataset bias in $A \rightarrow T$.

To measure model bias in $A \rightarrow T$ ($\Psi_A^M$), we trained an attacker function $f_A^{\hat{T}}$ that predicts model $M$'s task predictions ($\hat{T}$) using the true attribute label ($A$) as an input. If the attacker function $f_A^{\hat{T}}$ predicts $\hat{T}$ from input $A$ with high accuracy, it shows that true attribute label ($A$) has high predictability for $M$'s task predictions ($\hat{T}$), indicating a high model bias in $A \rightarrow T$ (refer to Figure 1).

For $A \rightarrow T$, we mathematically denote dataset bias ($\Psi_A^D$) and model bias ($\Psi_A^M$):

$$\Psi_A^D = Q(f_A^T(A), T) \qquad (1) \qquad\qquad \Psi_A^M = Q(f_A^{\hat{T}}(A), \hat{T}) \qquad (2)$$

Similarly, for $T \rightarrow A$, we mathematically denote dataset bias ($\Psi_T^D$) and model bias ($\Psi_T^M$):

$$\Psi_T^D = Q(f_T^A(T), A) \qquad (3) \qquad\qquad \Psi_T^M = Q(f_T^{\hat{A}}(T), \hat{A}) \qquad (4)$$

For $T \rightarrow A$, $f_T^A$ represents an attacker function that takes $T$ as input and predicts $A$, while $f_T^{\hat{A}}$ represents an attacker function that takes $T$ as input and predicts $\hat{A}$. We define bias amplification as the difference between the model bias and dataset bias, normalized by the sum of these biases. Using equations 1 and 2, and equations 3 and 4, we define bias amplification in both directions:

$$DPA_{A \rightarrow T} = \frac{\Psi_A^M - \Psi_A^D}{\Psi_A^M + \Psi_A^D + \epsilon} \qquad (5) \qquad\qquad DPA_{T \rightarrow A} = \frac{\Psi_T^M - \Psi_T^D}{\Psi_T^M + \Psi_T^D + \epsilon} \qquad (6)$$

Note, a small value $\epsilon \rightarrow 0^+$ is added in the denominator for numerical stability.

When measuring dataset or model bias in either direction, note that the input to the attacker function is the same (true attribute label $A$ for equations 1 and 2, and true task label $T$ for equations 3 and 4). This is because we followed Wang et al.'s [3] definition of directionality, where bias amplification

(change in biases) should be measured with respect to a fixed prior variable ($A$ or $T$ in this case). We offer a detailed proof of how $DPA$ follows Wang et al.'s definition of directionality in Section B.

Note that the model $M$'s predictions are not $100\%$ accurate and may contain errors. For example, assume we are measuring bias amplification in $A \to T$, and model $M$ achieves $70\%$ accuracy on task predictions $\hat{T}$. Since the true task labels $T$ are $100\%$ accurate (as they represent ground-truth), this $30\%$ gap in accuracy could influence bias amplification values. To prevent conflating prediction errors with bias, we adopt the procedure of Wang et al. [7] to equalize the dataset and model accuracy. In this case, we randomly flip $30\%$ of the true task labels $T$ so that their accuracy aligns with the model $M$'s $70\%$. As the bias in $T$ (or $A$ for $T \to A$) can vary significantly between two iterations of random flips, we measured bias amplification using confidence intervals across multiple iterations.

## 3.2 Benefits over existing predictability-based metrics

In addition to directionality, $DPA$ has these benefits over other predictability-based metrics like $LA$:

**Reports bias amplification in a fixed range** For any quality function $Q$ (such that its range is $[0, \infty)$ or $[0, \mathbb{R}^+]$), the range of $DPA$ is restricted to $[-1, 1]$. Since $DPA$ reports bounded values, it is easy to compare bias amplification across different models.

Note: While selecting $Q$, users must ensure that $0$ represents the worst possible performance by the attacker (i.e., low predictability or no bias), and the upper bound for $Q$ represents the best possible performance by the attacker (i.e., high predictability or significant bias). This is true for most typical choices for quality functions, such as accuracy or F1 score, but not for certain losses like cross-entropy. For cross-entropy, a lower value would indicate good performance by the attacker (i.e., high predictability or significant bias). To use cross-entropy (or similar loss functions) as our $Q$, we should modify it to $(1/\text{cross-entropy})$ or $(1 - \text{cross-entropy})$.

**Measures Relative Amplification** $DPA$ accounts for dataset bias by measuring the relative change in bias (or predictability), rather than the absolute change as done by $LA$. Assume we have an unbiased dataset $D_1$ with a dataset bias of $0$, and a highly biased dataset $D_2$ with a dataset bias of $0.9$. We train identical models $M_1$ and $M_2$ on $D_1$ and $D_2$, respectively. Assume the resulting model biases are $0.05$ for $M_1$ and $0.95$ for $M_2$. If we measure absolute change like $LA$, both $D_1$ and $D_2$ show the same increase in bias ($0.05$), making the two scenarios appear equally problematic. In practical settings, a model introducing bias in an originally unbiased dataset ($D_1$) is more concerning than one adding the same amount of bias to an already biased dataset ($D_2$). If we measure relative change like $DPA$, we can distinguish between the two scenarios — the relative increase in bias (or amplification) for $D_1$ is $1$, while for $D_2$ it is only about $0.06$. To further show how relative change in $DPA$ accounts for dataset bias, refer to the analysis we conducted in Section C.

**Robust to the choice of attacker function** Predictability-based metrics are highly sensitive to the hyperparameters of the attacker function used, as shown in [12]. Since $DPA$ measures relative change in predictability, it is more robust to different hyperparameters of the attacker function. In section D, we performed experiments on the ImSitu dataset and a controlled synthetic dataset to show how measuring relative change in predictability improves $DPA$'s robustness to attacker function.

## 4 Experiments and Results

To show that $DPA$ is the most reliable metric to measure bias amplification in classification datasets, we conducted experiments on one tabular dataset (COMPAS [13]) and two image datasets (COCO [14] and ImSitu [1]). For COCO and ImSitu, we used gender as the protected attribute. In this section, we describe the experimental details and results for these datasets. For the COMPAS dataset, we used race as the protected attribute. The experiment setup and results for COMPAS are in Section E.

## 4.1 COCO Experiment Setup

COCO [14] is an image dataset where each image has one or more people (male or female) along with various objects (e.g., ball, book). We performed an experiment similar to Wang et al. [3], to test whether different bias amplification metrics can correctly identify the direction in which a model amplifies bias. We gradually increased the $T \to A$ bias in COCO through image masking and evaluated which metrics were able to detect this increasing bias.

We used the gender-annotated version of the COCO dataset released by Wang et al. [7]. Each image is labeled with gender ($A$) and object ($T$) annotations. $A$ holds a single value: $\{\texttt{Female} : 0, \texttt{Male} : 1\}$. If the image has multiple people, $A$ corresponds to the gender of the person in the foreground. $T$ is a one-dimensional vector of 78 binary values: $[t_0, t_1, ..., t_{77}]$ (e.g., teddy bear, skateboard). $t_i = 1$ if at least one instance of that $i^{th}$ object is present in the image, else $t_i = 0$. We sampled balanced and unbalanced sub-datasets from COCO. The balanced dataset is subject to the constraint in Equation 7.

$$\forall y : \#(m, y) = \#(f, y) \tag{7}$$

Where $\#(m, y)$ represents the number of images where a male performs task $y$ and $\#(f, y)$ represents the number of images where a female performs task $y$. As these constraints are hard to satisfy, only a subset of 12 objects (or tasks) was used in the balanced sub-dataset. This resulted in 6156 images in the sub-dataset (3078 male and 3078 female images). We used the same 12 objects for the unbalanced sub-dataset, but relaxed the constraint from Equation 7 as shown in Equation 8. This resulted in a sub-dataset of 15743 images (8885 male and 6588 female images)

$$\forall y : \frac{1}{3} < \frac{\#(m, y)}{\#(f, y)} < 3 \tag{8}$$

To gradually increase $T \rightarrow A$ bias, we created four versions of each sub-dataset: (1) the original sub-dataset, and three other versions where the person in the image is progressively masked by (2) partially masking the segment, (3) completely masking the segment, and (4) completely the masking bounding box. As we progressively mask more of the person, the model loses access to visual gender cues and is forced to rely heavily on surrounding objects (tasks) to predict gender. This causes the $T \rightarrow A$ bias in the model to gradually increase. We evaluated whether bias amplification metrics report a steady increase in $T \rightarrow A$ scores as the $T \rightarrow A$ model bias increases across the four versions. Since all models are trained on COCO, the dataset bias remains constant. We can directly compare changes in amplification scores with changes in model bias.

We used the image masking annotations for COCO from [3]. We used two models: ViT_b_16 [15] and VGG16 [16] (both pretrained on ImageNet-1K [17]). We replaced the last fully connected layer of these pre-trained models with a layer of size $|T|$. We fine-tuned these models for 12 epochs on 8 dataset versions (4 versions of the balanced and 4 versions of the unbalanced sub-dataset). We used only two models as fine-tuning a model on 8 datasets is computationally expensive.

We measured bias amplification using co-occurrence-based metrics like $BA_{\rightarrow}$, $BA_{MALS}$, $Multi_{\rightarrow}$, and predictability-based metrics like $LA$ and $DPA$ (ours). For $LA$ and $DPA$, we used an MLP with two hidden layers as our attacker function. Additional experiment details are in Section J.

### 4.2 COCO Results

To quantify the $T \rightarrow A$ model bias in the ViT (or VGG), we computed a feature attribution score using Gradient Shap [18]. The attribution score measures the percentage contribution of non-person pixels in the ViT's prediction of the person's gender. A higher attribution score means the ViT is relying more on the background objects rather than the person to predict gender. In Table 2, we show the average attribution score for the ViT model, along with bias amplification values from different metrics. Row 2 reports results on various masked versions of the unbalanced COCO sub-dataset, while row 3 reports results for the balanced sub-dataset.

In the unbalanced sub-dataset, the attribution score increased with each masked version. This confirms that as the person was progressively masked, the ViT increasingly used background objects to predict gender, indicating a rise in $T \rightarrow A$ model bias. Only $DPA$ and $BA_{\rightarrow}$ reported increasing $T \rightarrow A$ scores, correctly capturing the growing model bias. Since metrics like $BA_{MALS}$ and $LA$ lack directionality, they failed to capture the increasing $T \rightarrow A$ model bias. Although $Multi_{\rightarrow}$ is directional, it too failed to capture the increasing $T \rightarrow A$ bias. This is because $Multi_{\rightarrow}$ cannot capture negative bias amplification scenarios (we have clearly demonstrated this using the COMPAS experiment in Section E).

In the balanced COCO sub-dataset, labeled background objects (i.e., task objects) are balanced with gender, so they should not provide predictive signals for gender. However, we observed a rising attribution score with each masked version. This indicates that the model used background cues that were not part of the labeled object (or task) set.

Table 2: $T \to A$ bias amplification as the person is progressively masked in the unbalanced and balanced COCO sub-datasets. Attribution scores indicate the $T \to A$ model bias in the ViT_b_16 model. For the unbalanced sub-dataset, $DPA$ and $BA_\to$ captured the increasing $T \to A$ model bias. For the balanced sub-dataset, only $DPA$ captured the increasing $T \to A$ model bias. Subscript shows confidence intervals.

| Dataset Split | Metric | Original | Partial Masked | Segment Masked | Bounding-Box Masked |
|---|---|---|---|---|---|
| | Image |  |  |  |  |
| | Attribution Map |  |  |  |  |
| Unbalanced | Attribution | $0.3827_{0.0026}$ | $0.4327_{0.0023}$ | $0.4461_{0.0020}$ | $0.5247_{0.0022}$ |
| | $DPA$ (ours) | $-0.0152_{0.0017}$ | $0.1365_{0.0013}$ | $0.6015_{0.0006}$ | $0.8085_{0.0003}$ |
| | $BA_\to$ | $-0.0227_{0.0001}$ | $0.0097_{0.0002}$ | $0.0188_{0.0003}$ | $0.0601_{0.0002}$ |
| | $Multi_\to$ | $0.1506_{0.0002}$ | $0.1179_{0.0004}$ | $0.3606_{0.0003}$ | $0.5607_{0.0010}$ |
| | $LA$ | $0.0368_{0.0052}$ | $0.0715_{0.0011}$ | $0.0942_{0.0020}$ | $0.0265_{0.0008}$ |
| | $BA_{MALS}$ | $0.0001_{0.0000}$ | $0.0004_{0.0000}$ | $-0.0016_{0.0000}$ | $0.0015_{0.0000}$ |
| Balanced | Attribution | $0.4568_{0.0026}$ | $0.4684_{0.0026}$ | $0.4834_{0.0026}$ | $0.4933_{0.0025}$ |
| | $DPA$ (ours) | $0.0230_{0.0008}$ | $0.0280_{0.0003}$ | $0.0380_{0.0030}$ | $0.1039_{0.0006}$ |
| | $BA_\to$ | $0.0000_{0.0000}$ | $0.0000_{0.0000}$ | $0.0000_{0.0000}$ | $0.0000_{0.0000}$ |
| | $Multi_\to$ | $0.0008_{0.0000}$ | $0.0010_{0.0000}$ | $0.0038_{0.0000}$ | $0.0043_{0.0000}$ |
| | $LA$ | $0.0006_{0.0002}$ | $0.0028_{0.0002}$ | $0.0020_{0.0005}$ | $0.0019_{0.0003}$ |
| | $BA_{MALS}$ | $0.0000_{0.0000}$ | $0.0000_{0.0000}$ | $0.0000_{0.0000}$ | $0.0000_{0.0000}$ |

To investigate this further, we visualized attribution maps for an image from the balanced COCO sub-dataset, as shown in Table 2. As the person was gradually masked, the ViT model increasingly focused on unlabeled objects, such as skis and ski poles, to predict gender. This shows that even if datasets are balanced, models can amplify bias from unlabeled objects at test time. This behavior is captured by $DPA$ as it reports increasing $T \to A$ scores. While $Multi_\to$ showed the same trend as DPA, it is not reliable as it does not capture negative bias amplification (as mentioned earlier). $BA_\to$ and $BA_{MALS}$ incorrectly assume that a balanced dataset contains no bias. They failed to capture the increasing $T \to A$ model bias and consistently reported zero bias amplification.

While the attribution maps in Table 2 show the ViT's reliance on unlabeled objects, they may not fully convince readers that balanced datasets can amplify biases from unlabeled objects. To make this point clear, we designed a controlled experiment (Section F) where we introduced a hidden $A \to T$ bias into the balanced COCO sub-dataset using one-pixel shortcuts (OPS) [19]. This setup mimics the presence of bias from unlabeled objects. Only $DPA$ was able to detect the bias amplification caused by this hidden bias. To further convince readers, we conducted another controlled experiment on hidden biases using the CMNIST dataset in Section G.

The VGG model showed similar trends to the ViT model on both the unbalanced and balanced COCO sub-datasets. We showed VGG results in Section H.

## 4.3 ImSitu Experiment Setup

ImSitu [1] is an image dataset where a person (male or female) performs an activity (e.g., playing, eating). We examined which metrics can correctly measure directional bias amplification in ImSitu. ImSitu did not provide image mask annotations, so we were unable to perform progressive person masking to verify directionality. Instead, we examined which metrics could detect $A \to T$ and $T \to A$ bias amplification in the original (unmasked) dataset.

We used the gender-annotated version of the ImSitu dataset released by Wang et al. [7]. Each image has gender ($A$) and activity ($T$) annotations. $A$ holds a single value: $\{\texttt{Female}: 0, \texttt{Male}: 1\}$. If the image has multiple people, $A$ corresponds to the gender of the person performing the activity. $T$ is a one-dimensional vector of 211 activities: $[t_0, t_1, ..., t_{210}]$ (e.g., repairing, curling). Each image has only one activity annotation — that activity is assigned 1 and the rest are 0.

We sampled unbalanced and balanced sub-datasets from ImSitu. To sample the balanced sub-dataset, we used the constraint in Equation 7. This resulted in a sub-dataset of 14600 images (7300 male and 7300 female images). For the unbalanced sub-dataset, we used a modified constrained (Equation 9). This resulted in a sub-dataset of 24301 images (14199 male and 10102 female images). We selected a subset of 205 activities (tasks) for both sub-datasets.

$$\forall y : \frac{1}{3} < \frac{\#(m,y)}{\#(f,y)} < 3 \tag{9}$$

The ImSitu experiment was much less expensive than the COCO masking experiment, as we had to train each model only on 2 sub-datasets. Along with ViT_b_16 [15] and VGG16 [16] (used in the COCO setup), we evaluated seven more models — MaxViT [20], [15], ViT_b_32 [15], SqueezeNet [21], Wide ResNet50 [22], Wide ResNet101 [22], MobileNet V2 [23], and Swin Tiny [24]. For each of the sub-datasets, we measured bias amplification caused by these models in two directions: bias amplification caused by gender ($A$) on activities ($T$): $A \rightarrow T$, and the bias amplification caused by activities ($T$) on gender ($A$): $T \rightarrow A$.

As mentioned in Section 4.1, since all models are trained on the same dataset (in this case, ImSitu), bias amplification only depends on model bias. We examined which bias amplification metrics correctly captured $A \rightarrow T$ and $T \rightarrow A$ model biases. We used co-occurrence-based metrics like $BA_\rightarrow$, $BA_{MALS}$, $Multi_\rightarrow$, and predictability-based metrics like $LA$ and $DPA$ (ours). Similar to COCO, for $LA$ and $DPA$, we used an MLP with two hidden layers as our attacker model. Additional experiment details are in Section J.

## 4.4 ImSitu Results

To quantify $A \rightarrow T$ and $T \rightarrow A$ model bias in ImSitu, we could not use feature attribution scores as ImSitu lacks person mask annotations. Instead, we used "conceptual sensitivity" scores based on Concept Activation Vectors (CAVs) [25]. Sensitivity score ($Sen$) quantifies how much a model's prediction is influenced by a specific concept.

We define $Sen_{A \rightarrow T}$ as the influence of the gender concept (male or female) in the overall task predictions. We define $Sen_{T \rightarrow A}$ as the contribution of task concepts (activities like playing, eating) in the overall gender predictions. If $Sen_{A \rightarrow T}$ is high, it means that the concept "male" (or female) highly influences the model's task predictions, indicating a strong $A \rightarrow T$ model bias. If $Sen_{T \rightarrow A}$ is high, it means that the task concepts (e.g., eating) highly influence the model's attribute predictions, indicating a strong $T \rightarrow A$ model bias.

In Table 3, we showed the sensitivity and bias amplification scores in the $A \rightarrow T$ direction for the unbalanced sub-dataset. In Table 4, we showed corresponding results for the balanced sub-dataset. For both tables, we ranked all models (from 1 to 9) by their $Sen_{A \rightarrow T}$ scores, from least biased to most biased. We checked which bias amplification metrics produced rankings closest to this model bias ranking. This allowed us to see which metrics best captured changes in model bias.

In the unbalanced case (Table 3), $DPA$'s ranking was very close to the model bias ranking — it differed by only one position (rank 5 vs. 6). All other metrics showed significant mismatches. In the balanced case (Table 4), DPA fully matched the $Sen_{A \rightarrow T}$ rankings. Other metrics, like $LA$ and $Multi_\rightarrow$, differed significantly. Metrics like $BA_\rightarrow$ and $BA_{MALS}$ reported zero bias amplification because they assume that a balanced dataset has no bias. We observed similar trends for the unbalanced and balanced sub-datasets in $T \rightarrow A$. The $T \rightarrow A$ results are shown in Section I.

Our experiments on COMPAS, COCO, and ImSitu show that only DPA reliably captures the three properties of a bias amplification metric: directionality, accurately measuring both positive and negative bias amplification, and working with both unbalanced and balanced datasets. No other metrics satisfy all three properties. $Multi_\rightarrow$ cannot measure negative bias amplification. $BA_\rightarrow$ and $BA_{MALS}$ always report zero amplification on balanced datasets. $LA$ cannot capture the direction in which a model amplifies biases. We also ran a controlled experiment on the directional metrics ($DPA$, $BA_\rightarrow$, $Multi_\rightarrow$) in Section K to show how $BA_\rightarrow$ fails to measure bias amplification in balanced datasets, and how $Multi_\rightarrow$ fails to capture negative bias amplification scenarios.

Table 3: $A \rightarrow T$ results for the ImSitu unbalanced sub-dataset. Models are shown in increasing order of model bias, indicated by their $Sen_{A\rightarrow T}$ scores. For each metric, the number in brackets shows the rank it assigned to that model. Rank 1 is the lowest (most negative) bias amplification, and rank 9 is the highest (most positive). Red numbers show where the metric's rank differs from the $Sen_{A\rightarrow T}$ rank. $DPA$ is the most reliable metric, as its model rankings best match the $Sen_{A\rightarrow T}$ rankings. All values are scaled by 100.

| Models | $Sen_{A\rightarrow T}$ | $DPA_{A\rightarrow T}$ | LA | $BA_{\rightarrow}$ | $Multi_{\rightarrow}$ | $BA_{MALS}$ |
|---|---|---|---|---|---|---|
| MaxViT [20] | 0.017 (1) | −0.147 (1) | 0.130 (4) | −0.157 (3) | 0.553 (6) | 0.211 (6) |
| ViT_b_32 [15] | 0.018 (2) | −0.050 (2) | 0.209 (6) | −0.146 (5) | 0.606 (8) | 0.248 (8) |
| ViT_b_16 [15] | 0.025 (3) | −0.036 (3) | 0.195 (5) | −0.145 (7) | 0.568 (7) | 0.228 (7) |
| SqueezeNet [21] | 0.082 (4) | −0.022 (4) | 0.009 (1) | −0.172 (1) | 48.40 (9) | 25.58 (9) |
| Wide ResNet101 [22] | 0.144 (5) | 0.064 (6) | 0.281 (9) | −0.146 (5) | 0.348 (4) | 0.109 (4) |
| MobileNet V2 [23] | 0.156 (6) | −0.011 (5) | 0.087 (3) | −0.158 (2) | 0.505 (5) | 0.193 (5) |
| Wide ResNet50 [22] | 0.190 (7) | 0.118 (7) | 0.253 (8) | −0.141 (8) | 0.260 (3) | 0.061 (3) |
| Swin Tiny [24] | 0.225 (8) | 0.318 (8) | 0.064 (2) | −0.151 (4) | 0.159 (2) | −0.074 (1) |
| VGG16 [16] | 0.272 (9) | 0.329 (9) | 0.210 (7) | −0.051 (9) | 0.097 (1) | −0.045 (2) |

Table 4: $A \rightarrow T$ results for the ImSitu balanced sub-dataset. This table follows the same setup as Table 3. Red numbers show where the metric's rank differs from the $Sen_{A\rightarrow T}$ rank. $DPA$ is the most reliable metric, as its model rankings exactly match the $Sen_{A\rightarrow T}$ rankings.

| Models | $Sen_{A\rightarrow T}$ | $DPA_{A\rightarrow T}$ | LA | $BA_{\rightarrow}$ | $Multi_{\rightarrow}$ | $BA_{MALS}$ |
|---|---|---|---|---|---|---|
| SqueezeNet [21] | 0.000 (1) | 0.003 (1) | 0.003 (8) | 0.000 (1) | 48.47 (9) | 0.000 (1) |
| Wide ResNet50 [22] | 0.089 (2) | 0.085 (2) | 0.002 (7) | 0.000 (1) | 0.705 (3) | 0.000 (1) |
| Wide ResNet101 [22] | 0.098 (3) | 0.094 (3) | 0.000 (3) | 0.000 (1) | 0.730(4) | 0.000 (1) |
| MobileNet V2 [23] | 0.112 (4) | 0.256 (4) | 0.007 (9) | 0.000 (1) | 1.379 (5) | 0.000 (1) |
| Swin Tiny [24] | 0.211 (5) | 0.282 (5) | 0.001 (5) | 0.000 (1) | 0.038 (1) | 0.000 (1) |
| ViT_b_32 [15] | 0.283 (6) | 0.285 (6) | 0.001 (5) | 0.000 (1) | 1.510 (6) | 0.000 (1) |
| ViT_b_16 [15] | 0.302 (7) | 0.302 (7) | 0.000 (2) | 0.000 (1) | 1.599 (7) | 0.000 (1) |
| MaxViT [20] | 0.478 (8) | 0.326 (8) | −0.002 (1) | 0.000 (1) | 1.762 (8) | 0.000 (1) |
| VGG16 [16] | 0.497 (9) | 0.341 (9) | 0.000 (3) | 0.000 (1) | 0.073 (2) | 0.000 (1) |

## 5 Discussion

### 5.1 Why and when should we use DPA?

Table 5: Example dataset and model prediction tables to demonstrate when we should use $BA_{\rightarrow}$ and DPA. We used the same tables for the hiring scenario and the indoor-outdoor scenario.

(a) Original Dataset

| | $T = 0$ | $T = 1$ |
|---|---|---|
| $A = 0$ | 50 | 150 |
| $A = 1$ | 22 | 78 |

(b) Model Predictions

| | $\hat{T} = 0$ | $\hat{T} = 1$ |
|---|---|---|
| $A = 0$ | 15 | 185 |
| $A = 1$ | 9 | 91 |

In real-world machine learning applications, it is common to train multiple models on the same dataset to determine which model performs best. Often, these models achieve similar accuracies, making model selection difficult — a phenomenon known as the Rashomon effect [26, 27]. In such cases, metrics like $DPA$ can offer valuable insights into model selection by highlighting which models preserve, reduce, or worsen existing biases.

Based on our experiments, we found $DPA$ to be the most reliable metric to measure bias amplification. However, there are cases where other directional metrics like $BA_{\rightarrow}$ could be more appropriate. Let us take two scenarios where we measure $A \rightarrow T$ bias amplification in $DPA$ and $BA_{\rightarrow}$. Consider a hiring dataset where 200 men ($A = 0$) and 100 women ($A = 1$) apply for a job. Out of these, 50 men and 22 women are hired ($T = 0$), while the rest are rejected ($T = 1$) — refer to Table 5a. Assume we train a model (M) on this dataset, which gives predictions as shown in Table 5b.

According to $DPA$, the $A \rightarrow T$ bias amplification is large. This is because for both men and women, $\hat{T}$ is much more unbalanced than $T$. For men: $T$ ratio $= 3:1(150/50)$, $\hat{T}$ ratio $= 12.3:1(185/15)$. For women: $T$ ratio $= 3.5:1(78/22)$, $\hat{T}$ ratio $= 10.1:1(91/9)$. According to $BA_{\rightarrow}$, bias amplification is small. This is because the ratio of hired men and women is similar in the dataset and in the model's predictions. Hired men and women in the dataset: 25% (50/200) and 22% (22/100), respectively. Hired men and women in the model's predictions: 7.5% (15/200) and 9% (9/100), respectively. Our goal is to ensure men and women have a similar acceptance ratio. Since the acceptance ratio between

men and women is similar in the dataset and in the model's predictions, the small bias amplification reported by $BA_\rightarrow$ is appropriate. In scenarios where we desire equal opportunity (e.g., hiring datasets where men and women should have equal acceptance ratios), using $BA_\rightarrow$ is better.

In another scenario, consider a dataset of men ($A = 0$) and women ($A = 1$), where each person is either indoors ($T = 0$) or outdoors ($T = 1$). Assume the same tables as before (Tables 5a and 5b). Here, our goal is to ensure that we have a balanced count of men and women, indoors and outdoors. We should report high bias amplification as $\hat{T}$ is highly unbalanced with respect to $T$ (for both men and women). In this case, the large bias amplification reported by $DPA$ is appropriate. $BA_\rightarrow$ and $DPA$ capture different notions of bias, and the right choice of metric depends on the type of bias we seek to address.

## 5.2 Limitations and Future Work

Predictability-based metrics like DPA (and LA) rely on training attacker models to estimate bias amplification. These attackers can be sensitive to training instabilities — such as getting stuck in local minima. This may lead to inconsistent results. To ensure stable results, we must run the attacker model multiple times, which increases computational cost. In contrast, co-occurrence-based metrics like $BA_\rightarrow$ or $Multi_\rightarrow$ are cheaper to compute. We have compared the runtime for all bias amplification metrics in Section L.

$DPA$ cannot measure bias amplification at the level of individual $A - T$ pairs — the bias amplification caused by a specific $A$ (e.g., women) on a specific $T$ (e.g., cooking), or vice versa. Instead, it measures overall $A \rightarrow T$ and $T \rightarrow A$ bias amplification, aggregated across all $A$ and $T$. Suppose a model amplifies bias on the cooking task when women are present, but reduces bias on the gardening task when men are present. $DPA$ would report only the net effect across both $A$'s and both $T$'s, masking these opposing trends.

As part of future work, we aim to disaggregate bias amplification into individual $A - T$ pairs. We can use techniques like feature attribution to find out the influence of a specific attribute $A = A_i$ (e.g., male), in predicting a specific task $T = T_j$ (e.g., cooking). This attribution value can be used as a weak proxy to measure dataset or model bias between $A_i, T_j$. Causal attacker models or interpretable attackers are other potential approaches to disaggregate bias amplification into individual $A - T$ pairs. This finer-grained view can support targeted bias intervention strategies.

## 5.3 Conclusion

In this work, we introduced $DPA$, a one-stop predictability-based metric to measure bias amplification in classification problems. $DPA$ is the only metric that can (1) accurately identify positive and negative bias amplification scenarios, offering valuable insights into model selection by highlighting which models preserve, worsen, or reduce data bias, (2) identify the source of bias amplification as the metric is directional (3) measure bias amplification for both unbalanced and balanced datasets. $DPA$ eliminates the need to evaluate models on multiple bias amplification metrics to verify each of the three aspects of bias amplification. $DPA$ is also more robust and easier to interpret than current predictability-based metrics like $LA$. While $DPA$ is generally the most reliable metric to measure bias amplification, there are bias scenarios (e.g., hiring datasets where rejections are almost always more than acceptances) where other metrics like $BA_\rightarrow$ may be more suitable. Before using $DPA$, we must have an accurate understanding of the type of bias we want to address.

## 6 Acknowledgments

We acknowledge the Research Computing at Arizona State University for providing HPC resources [28] that have contributed to the results reported in this paper. We also thank Mirali Purohit for her helpful suggestions throughout the development of this work.

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

## A  $Multi_\rightarrow$ **explanation**

To understand why $Multi_\rightarrow$ cannot differentiate between positive bias amplification and negative bias amplification (i.e., bias reduction), let us take a look at its formulation.

$$Multi_\rightarrow = X, Var(\Delta_{gm})$$

$$X = \frac{1}{|\mathcal{G}|\,|\mathcal{M}|} \sum_{g \in \mathcal{G}} \sum_{m \in \mathcal{M}} y_{gm} |\Delta_{gm}| + (1 - y_{gm}) |-\Delta_{gm}| \tag{10}$$

where,

$$y_{gm} = 1[P(m = 1, g = 1) > P(g = 1)P(m = 1)]$$

and,

$$\Delta_{gm} = \begin{cases} P(\hat{g} = 1 | m = 1) - P(g = 1 | m = 1) \\ \text{if measuring } M \rightarrow G \\ P(\hat{m} = 1 | g = 1) - P(m = 1 | g = 1) \\ \text{if measuring } G \rightarrow M \end{cases} \tag{11}$$

Following [4], $M$ represents the attribute groups, and $G$ represents the task groups.

From Equation 10, we get

$$X = \frac{1}{|\mathcal{G}|\,|\mathcal{M}|} \sum_{g \in \mathcal{G}} \sum_{m \in \mathcal{M}} y_{gm} |\Delta_{gm}| + |\Delta_{gm}| - y_{gm} |\Delta_{gm}|$$

$$\implies X = \frac{1}{|\mathcal{G}|\,|\mathcal{M}|} \sum_{g \in \mathcal{G}} \sum_{m \in \mathcal{M}} |\Delta_{gm}| \tag{12}$$

Hence, we see from Equations 11 and 12 that $Multi_\rightarrow$ simply measures the average absolute differences for the conditional probabilities. Due to the absolute term, $Multi_\rightarrow$ cannot capture negative bias amplification.

## B  **Explaining Directionality**

To prove that $DPA$ follows Wang et al.'s definition of directionality [3], let us see how $DPA$ measures bias amplification in $A \rightarrow T$. As per equation 1, to calculate dataset bias ($\Psi_A^D$), the attacker function $f_A^T$ tries to model the relationship of $P(T|A)$. For $Q$ function of accuracy, we can rewrite the equation 1 as:

$$\Psi_A^D = Q(f_A^T(A), T) = \sum_{A_i \in A} P(A_i)P(T_{mi}|A_i) \tag{13}$$

where $T_{mi}$ is such that $P(T_{mi}|A) \geq P(T_k|A_i) \forall k$.

Similarly, we can rewrite equation 2 as:

$$\Psi_A^M = Q(f_A^{\hat{T}}(A), \hat{T}) = \sum_{A_i \in A} P(A_i)P(\hat{T}_{mi}|A_i) \tag{14}$$

Then $DPA_{A \rightarrow T}$ can be defined as:

$$DPA_{A \rightarrow T} = \sum_{A_i \in A} P(A_i)(P(\hat{T}_{mi}|A_i) - P(T_{mi}|A_i)) \tag{15}$$

Similarly, for $T \rightarrow A$ direction we can say:

$$DPA_{T \rightarrow A} = \sum_{T_i \in T} P(T_i)(P(\hat{A}_{mi}|T_i) - P(A_{mi}|T_i)) \tag{16}$$

Let us compare this to Wang et al.'s definition of directionality [3]. They defined their metric $BA_\rightarrow$ in the following manner:

$$BA_\rightarrow = \frac{1}{|A||T|} \sum_{a \in A, t \in T} y_{at}\Delta_{at} + (1 - y_{at})(-\Delta_{at}) \tag{17}$$

where,

$$y_{at} = 1[P(A_a = 1, T_t = 1) > P(A_a = 1)P(T_t = 1)] \tag{18}$$

$$\Delta_{at} = \begin{cases} P(\hat{T}_t = 1 | A_a = 1) - P(T_t = 1 | A_a = 1) \\ \text{if measuring } A \rightarrow T \\ P(\hat{A}_a = 1 | T_t = 1) - P(A_a = 1 | T_t = 1) \\ \text{if measuring } T \rightarrow A \end{cases} \tag{19}$$

For $T \rightarrow A$, $BA_\rightarrow$ measures the change in $P(\hat{A}|T)$ with respect to $P(A|T)$, i.e., change in the conditional probability of $\hat{A}$ vs. $A$ with respect to a fixed prior $T$. Similarly, for $A \rightarrow T$, $BA_\rightarrow$ measures change in the conditional probability of $\hat{T}_{mi}$ vs. $T_{mi}$ with respect to a fixed prior $A_i$.

Thus, we see both $BA_\rightarrow$ and $DPA$ are proportional to the similar changes in conditional probabilities with fixed priors. Therefore, $DPA$ follows the same concept of directionality as $BA_\rightarrow$.

## C   Relative vs. Absolute Change in Predictability

To further show how relative change in $DPA$ accounts for dataset bias, we plotted the relationship between $LA$ and model bias ($\lambda^M$) in Figure 2a, and between $DPA$ and model bias ($\Psi^M$) in Figure 2b, across three values of dataset bias — 0.1, 1, and 2 (dataset bias is denoted with $\Psi^D$ for $DPA$ and $\lambda^D$ for $LA$ ). We observed that the slope of the $LA$ vs. $\lambda^M$ curve remains constant regardless of the dataset bias ($\lambda^D$). In contrast, the $DPA$ vs. model bias ($\Psi^M$) curve shows steeper slopes for smaller dataset biases ($\Psi^D$) and flatter slopes for higher dataset biases. This means that for nearly unbiased datasets, $DPA$ reports high amplification even for small increases in model bias, whereas for highly biased datasets, it reports lower amplification for a similar increase in bias. This shows that, unlike absolute change, relative change (used in $DPA$) accounts for dataset bias.

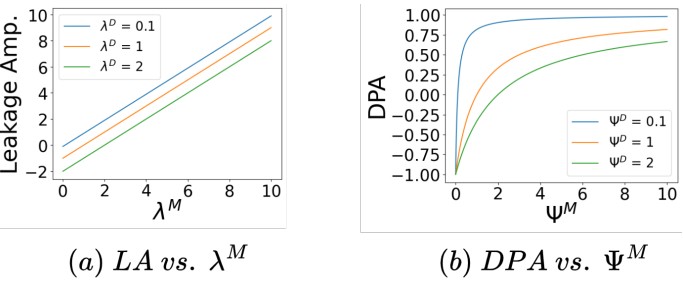

$(a)$ $LA$ vs. $\lambda^M$            $(b)$ $DPA$ vs. $\Psi^M$

Figure 2: The graphs show trends between (a) $LA$ vs model bias ($\lambda^M$), and between (b) $DPA$ vs model bias ($\Psi^M$), at different values of dataset bias ($\lambda^D$ for $LA$ and $\Psi^D$ for $DPA$). For the same model bias, $DPA$ reported much higher bias amplification values (compared to $LA$) when the dataset bias is small.

## D   Attacker Robustness

To show how measuring relative change in predictability improves $DPA$'s robustness to attacker function, we conduct an experiment using a controlled synthetic dataset and a real-world dataset like ImSitu [1].

### D.1   Synthetic Experiment

We define $A : \mathcal{N}(3, 2)$. We define $T$ and $\hat{T}$ in the following manner:

$$T = poly(A + (\alpha_1 * \epsilon), p) \tag{20}$$

$$\hat{T} = poly(A + (\alpha_2 * \epsilon), p) \qquad (21)$$

Here $poly(x, p)$ represents any $p^{th}$ degree polynomial of $x$ and $\epsilon : \mathcal{N}(0, 1)$. To demonstrate positive bias amplification, we want $\hat{T}$ to be a better predictor of $A$, compared to $T$. Hence, we set $\alpha_2 < \alpha_1$.

As the attacker is generally a simple polynomial function, we used a simple Fully Connected Network as the attacker. We parameterized the depths ($d$) and width ($w$) of the attacker function. This parameterization helped us create a large variety of attacker functions (each with a different depth and width). The parameters of the experiment are shown in Table 3. For each attacker, we used a combination of TanH and ReLU activations.

We created two versions of $DPA$ — (1) $DPA$ with absolute change in bias (or predictability), to mimic the behavior of current metrics like $LA$, (2) $DPA$ with relative change in bias, which is our formulation. For both versions, we used the inverse of RMSE loss as the quality function ($Q$).

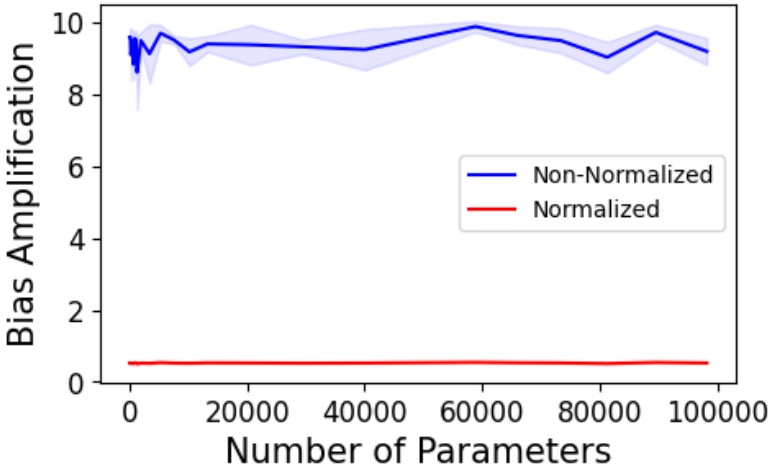

Figure 3: For different attacker functions, we show the bias amplification reported by $DPA$ with absolute change (blue line) and $DPA$ with relative change (red line) on the synthetic dataset.

For different attacker functions, we compared the values reported $DPA$ with absolute change (blue line) and by $DPA$ with relative change (red line) in Figure 3. $DPA$ with absolute change shows unstable bias amplification scores with high variance across attackers. $DPA$ with relative change shows a stable bias amplification score with minimal variance across attackers. This shows that $DPA$ with relative change is more robust to attacker function hyperparameters.

Table 6: Experiment Parameters

| Parameter | $p$ | $\alpha_1$ | $\alpha_2$ | $w$ | $d$ |
|-----------|-----|------------|------------|------------|--------|
| Value | 2 | 1 | 2 | $[20, 500]$ | $[2, 6]$ |

### D.2 ImSitu Experiment

We show that measuring relative change in predictability (as done in $DPA$) is robust to attacker hyperparameters on a real-world dataset like ImSitu. We used the same setup as the ImSitu experiment in section 4.3. We trained a ViT_b_16 model on the unbalanced ImSitu sub-dataset. We measured bias amplification using attackers with different hyperparameters.

We used a Fully Connected Network as our attacker function. We parameterized the width $w$ (in the range of $[5, 40]$ neurons) of the attacker function. This parameterization helped us create a large variety of attacker functions (each with a different width). For each attacker, we used ReLU activations. We used the Adam Optimizer with a learning rate of $5 \times 10^{-3}$.

We created two versions of $DPA$ — (1) $DPA$ with absolute change in bias (or predictability), to mimic the behavior of current metrics like $LA$, (2) $DPA$ with relative change in bias, which is our formulation. For both versions, we used the inverse of categorical cross-entropy loss as the quality function ($Q$).

For different attacker functions, we compared the values reported $DPA$ with absolute change (blue line) and by $DPA$ with relative change (red line) in Figure 4. $DPA$ with absolute change shows unstable bias amplification scores with high variance across attackers. $DPA$ with relative change shows a stable bias amplification score with minimal variance across attackers. This shows that even in real-world datasets like ImSitu, $DPA$ with relative change is more robust to attacker function hyperparameters.

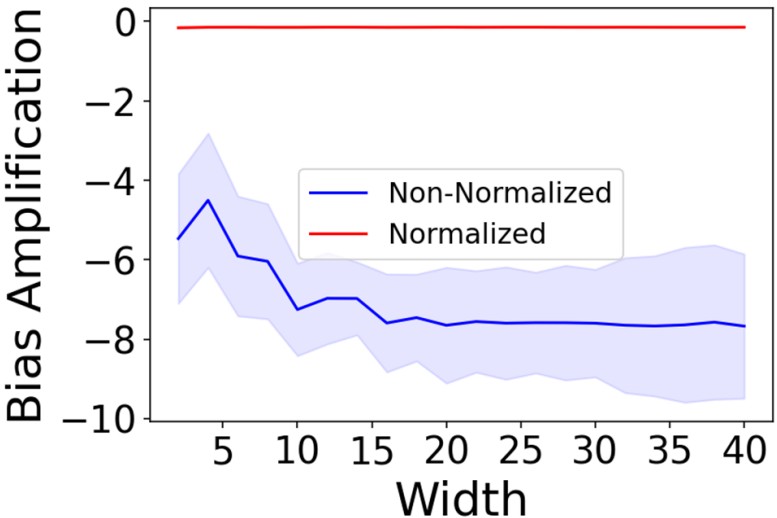

Figure 4: For different attacker functions, we show the bias amplification reported by $DPA$ with absolute change (blue line) and $DPA$ with relative change (red line) on the ImSitu dataset.

# E    COMPAS experiment

### E.1    Experiment Setup

COMPAS is a tabular dataset containing information about individuals who have been previously arrested. Each entry is associated with 52 features. We used five features: `age`, `juv_fel_count`, `juv_misd_count`, `juv_other_count`, `priors_count`.

We limited the dataset to 2 races (`Caucasian` or `African-American`), which we used as the protected attribute ($A$). The task ($T$) was recidivism (i.e., if the person was arrested again for a crime in the next 2 years). Hence, $A = \{$`Caucasian` $: 0,$ `African-American` $: 1\}$ and $T = \{$`No Recidivism` $: 0,$ `Recidivism` $: 1\}$.

We created balanced and unbalanced versions of the COMPAS dataset. For the unbalanced dataset, we sampled all available COMPAS instances (attributes, race labels, and recidivism labels) for each of the four $A$ and $T$ pairs. For the balanced dataset, we sampled an equal number of instances across the four $A$ and $T$ pairs. The counts for the $A$ and $T$ pairs in the unbalanced dataset are shown in the top-left quadrant of Table 7a, and for the balanced dataset, in the top-left quadrant of Table 7b.

We trained a decision tree model on the unbalanced and the balanced COMPAS datasets. Each model predicts a person's race ($\hat{A}$) and recidivism ($\hat{T}$) based on the 5 selected features. We measured the bias amplification caused by each model in two directions: bias amplification caused by race ($A$) on recidivism ($T$), referred to as $A \rightarrow T$, and the bias amplification caused by recidivism ($T$) on race ($A$), referred to as $T \rightarrow A$. In this experiment we only evaluated directional metrics – $DPA$ (ours), $BA_{\rightarrow}$ and $Multi_{\rightarrow}$.

The problem of mapping a binary variable ($A$) to another binary variable ($T$) for the $A \rightarrow T$ direction and vice-versa is trivial and can be done using a contingency matrix as well. Thus, we use two variations of $DPA$. For $DPA$ (ANN), we used a 3-layer dense neural network (with a hidden layer of size 4 and sigmoid activations) as the attacker function. For $DPA$ (MAT), we calculate the probabilities based on a contingency matrix instead of an attacker function. Note that quality equalization is still applied as a preprocessing step.

### E.2 Results

While interpreting COMPAS results, note that a co-occurrence-based metric like $BA_{\rightarrow}$ and a predictability-based metric like $DPA$ capture different notions of bias.

$BA_{\rightarrow}$ classifies each $A - T$ pair in the dataset as a majority or minority pair using equation 18. It only measures if the counts of the majority pair increased (positive bias amplification) or decreased (negative bias amplification) in the model predictions or vice versa.

$DPA$ does not select a majority or a minority $A - T$ pair. It measures the change in the task distribution given the attribute (and vice versa). For instance, if $A$ and $T$ are binary, $DPA$ measures if the absolute difference in counts between $T = 0$ and $T = 1$ increased (positive bias amplification) or decreased (negative bias amplification) in the model predictions. Both $BA_{\rightarrow}$ and $DPA$ offer different yet valuable insights into bias amplification.

Table 7: **COMPAS Dataset**: Counts of the protected attribute (race) and task (recidivism) in the dataset (represented as $A$ and $T$) and in the model predictions (represented as $\hat{A}$ and $\hat{T}$) for the balanced and unbalanced COMPAS set. Here: $A = \{$Caucasian $: 0,$ African-American $: 1\}$ and $T = \{$No Recidivism $: 0,$ Recidivism $: 1)\}$.

| | $A = 0$ | $A = 1$ | $\hat{A} = 0$ | $\hat{A} = 1$ |
|---|---|---|---|---|
| $T = 0$ | 1229 | 1402 | 1056 | 1575 |
| $T = 1$ | 874 | 1773 | 1115 | 1532 |
| $\hat{T} = 0$ | 1165 | 1546 | – | – |
| $\hat{T} = 1$ | 938 | 1629 | – | – |

(a) Unbalanced COMPAS Set

| | $A = 0$ | $A = 1$ | $\hat{A} = 0$ | $\hat{A} = 1$ |
|---|---|---|---|---|
| $T = 0$ | 874 | 874 | 1083 | 665 |
| $T = 1$ | 874 | 874 | 896 | 852 |
| $\hat{T} = 0$ | 1145 | 948 | – | – |
| $\hat{T} = 1$ | 603 | 800 | – | – |

(b) Balanced COMPAS Set

Table 8: **COMPAS Results**: The first two columns depict the bias amplification values for the unbalanced COMPAS set (Table 7a), while the last two columns depict the bias amplification values for the balanced COMPAS set.(Table 7b). Subscript shows confidence intervals.

| Method | Unbalanced | | Balanced | |
|---|---|---|---|---|
| | $T \rightarrow A$ | $A \rightarrow T$ | $T \rightarrow A$ | $A \rightarrow T$ |
| $BA_{\rightarrow}$ | $-0.078_{0.031}$ | $-0.038_{0.001}$ | $0.000_{0.000}$ | $0.000_{0.000}$ |
| $Multi_{\rightarrow}$ | $0.078_{0.026}$ | $0.038_{0.001}$ | $0.066_{0.007}$ | $0.099_{0.006}$ |
| $DPA$ (ANN) | $0.063_{0.005}$ | $-0.040_{0.002}$ | $0.061_{0.008}$ | $0.100_{0.004}$ |
| $DPA$ (MAT) | $0.022_{0.008}$ | $-0.036_{0.001}$ | $0.066_{0.004}$ | $0.098_{0.006}$ |

#### E.2.1 Unbalanced COMPAS dataset

The bias amplification scores for the unbalanced case are shown in the first two columns of Table 8.

$T \rightarrow A$: For $BA_{\rightarrow}$, when $T = 0$, the count of the majority class $A = 0$ decreased from 1229 in the dataset to 1056 in the model predictions. Similarly, when $T = 1$, the count of the majority class $A = 1$ decreased from 1773 in the dataset to 1532 in the model predictions. Since the count of the majority classes decreased in the model predictions, $BA_{\rightarrow}$ reported a negative bias amplification in $T \rightarrow A$.

For $DPA$, when $T = 0$, the difference in counts between $A = 0$ and $A = 1$ increased from 173 ($1402 - 1229 = 173$) in the dataset to 519 ($1575 - 1056 = 519$) in the model predictions. However, when $T = 1$, the difference in counts between $A = 0$ and $A = 1$ decreased from 899 ($1773 - 874 = 899$) in the dataset to 417 ($1532 - 1115 = 417$) in the model predictions. Since the decrease in bias when $T = 1$ is larger than the increase in bias when $T = 0$ ($899 - 417 > 519 - 173$), we might naively assume a negative bias amplification in $T \rightarrow A$.

This naive assumption overlooks the conflation of model errors and model biases (discussed in Section 3.1). Here, the decision tree model has a low accuracy when predicting $\hat{A}$ (approx. $69\%$); hence, $31\%$ of instances in $A$ are randomly flipped to match the model's accuracy. As a result, the biases in the perturbed $A$ are less than $\hat{A}$, indicating a positive bias amplification. The positive score reported by $DPA$ is not an incorrect result. It is the low model accuracy that misleadingly suggests a negative bias amplification. $Multi_\rightarrow$ reports positive bias amplification as it cannot capture negative bias amplification scenarios (discussed in Section A).

$A \rightarrow T$: For $BA_\rightarrow$, when $A = 0$, the count of the majority class $T = 0$ decreased from 1229 in the dataset to 1165 in the model predictions. Similarly, when $A = 1$, the count of the majority class $T = 1$ decreased from 1773 in the dataset to 1546 in the model predictions. Since the count of the majority classes decreased in the model predictions, $BA_\rightarrow$ reported negative bias amplification in $A \rightarrow T$.

For $DPA$, when $A = 0$, the difference in counts between $T = 0$ and $T = 1$ decreased from 355 $(1229 - 874 = 355)$ in the dataset to 227 $(1165 - 938 = 227)$ in the model predictions. Similarly, when $A = 1$, the difference in counts between $T = 0$ and $T = 1$ decreased from 371 $(1773 - 1402 = 371)$ to 83 $(1629 - 1546 = 83)$ in the model predictions. Since the overall count difference decreased in the model predictions, $DPA$ reported negative bias amplification in $A \rightarrow T$.

$Multi_\rightarrow$ reported positive bias amplification as it cannot capture negative bias amplification. It only measures the magnitude of bias amplification but not its sign.

### E.2.2 Balanced COMPAS Dataset

The bias amplification scores for the balanced case are shown in the last two columns of Table 8.

$T \rightarrow A$: Since $BA_\rightarrow$ assumes a balanced dataset is unbiased, $BA_\rightarrow$ reported zero bias amplification in $T \rightarrow A$. For $DPA$, when $T = 0$, the count difference between $A = 0$ and $A = 1$ increased from 0 $(874 - 874 = 0)$ in the dataset to 418 $(1083 - 665 = 418)$ in the model predictions. Similarly, when $T = 1$, the difference in counts between $A = 0$ and $A = 1$ increased from 0 $(874 - 874 = 0)$ in the dataset to 44 $(896 - 852 = 44)$ in the model predictions. Since the overall difference increased in the model predictions, $DPA$ reported positive bias amplification in $T \rightarrow A$.

$A \rightarrow T$: Since the dataset is balanced, $BA_\rightarrow$ reported zero bias amplification in $A \rightarrow T$. For $DPA$, when $A = 0$, the difference in counts between $T = 0$ and $T = 1$ increased from 0 $(874 - 874 = 0)$ in the dataset to 542 $(1145 - 603 = 542)$ in the model predictions. Similarly, when $A = 1$, the difference in counts between $T = 0$ and $T = 1$ increased from 0 $(874 - 874 = 0)$ in the dataset to 148 $(948 - 800 = 148)$ in the model predictions. Since the overall count difference increased in the model predictions, $DPA$ reported positive bias amplification in $A \rightarrow T$.

$Multi_\rightarrow$ reported positive bias amplification as it only looks at the magnitude of amplification scores.

## F  Simulating Biases in a Balanced COCO Dataset

Table 9: Bias Amplification scores for various metrics at different values of $L$ in the COCO dataset

| Test Accuracy (w/o shortcuts) | Test Accuracy (shortcuts) | L | $BA_\rightarrow$ | $Multi_\rightarrow$ | $DPA$ (ours) | $BA_{MALS}$ | $LA$ |
|---|---|---|---|---|---|---|---|
| 55.34% | 71.20% | 0.05 | $0.0000_{0.0000}$ | $0.2569_{0.0000}$ | $0.0829_{0.0007}$ | $0.2586_{0.0000}$ | $0.0000_{0.0000}$ |
| 55.34% | 72.58% | 0.10 | $0.0000_{0.0000}$ | $0.2707_{0.0000}$ | $0.0993_{0.0014}$ | $0.0032_{0.0000}$ | $0.0000_{0.0000}$ |
| 54.66% | 74.31% | 0.15 | $0.0000_{0.0000}$ | $0.2879_{0.0000}$ | $0.1098_{0.0009}$ | $0.1793_{0.0000}$ | $0.0000_{0.0000}$ |
| 54.50% | 75.01% | 0.20 | $0.0000_{0.0000}$ | $0.3017_{0.0000}$ | $0.1380_{0.0010}$ | $0.2448_{0.0000}$ | $0.0000_{0.0000}$ |

In this controlled experiment, we show that even if attributes and tasks are balanced in a dataset, models can pick up biases from unlabeled elements and amplify them at test time. We chose 2 out of the 12 task objects from the COCO balanced dataset used in Section 4.1: $A = \{\texttt{Female}: 0, \texttt{Male}: 1\}$ and $T = \{\texttt{Sink}: 0, \texttt{Not Sink}: 1\}$. Here, "Not Sink" refers to COCO images that do not have a sink. Our balanced dataset has 288 images for each $A$ and $T$ pair.

We created biases in our balanced dataset using one-pixel shortcuts (OPS) [19]. In OPS, a fixed pixel with position $(x_i, y_i)$ is assigned the same value in all images of a class. This makes it easier for a model to predict a class, as it only needs to learn that all images in that class share the same pixel. By applying OPS, we simulated how an unlabeled element (here, the shortcut pixel) could create biases in a balanced dataset.

We used two parameters, $\beta_1$ and $\beta_2$, where $0 \leq \beta_1, \beta_2 \leq 100$ and $\beta_1 > \beta_2$, to control how many shortcuts were added. For "sink" images, we applied OPS to $\beta_1\%$ images with males and $\beta_2\%$ images with females. For "not sink" images, we swapped the parameters, applying OPS to $\beta_2\%$ images with males and $\beta_1\%$ images with females.

Since $\beta_1 > \beta_2$, "sink" images with males had more shortcuts than those with females, while "not sink" images with females had more shortcuts than those with males. Although the dataset is balanced in terms of attributes and tasks, a model trained on this dataset is more likely to predict males for "sink" images and females for "not sink" images at test time. In essence, we introduced an $A \rightarrow T$ bias to simulate unlabeled bias elements in the balanced COCO dataset.

Similar to our experiment in Section 4.1, we trained a VGG16 model on this balanced COCO dataset. We trained the model for 15 epochs with a batch size of 32. We used the SGD optimizer, with a learning rate of $0.01$ and momentum of $0.5$. We used the binary cross-entropy loss for training.

Similar to the training dataset, we used 288 instances for each $A$ and $T$ pair in our test set. To verify if OPS introduced an $A \rightarrow T$ bias in the trained VGG16 model, we created two versions of our test set. In version 1, we did not introduce any shortcuts. In version 2, we applied shortcuts using parameters $\beta_1$ and $\beta_2$ as described above.

If the accuracy of VGG16 in version 2 of the test set (where shortcuts are applied) is greater than version 1 (where no shortcuts exist), we can confirm that the model has learned an $A \rightarrow T$ bias. This is because if the VGG16 model has learned the shortcuts (or biases) we introduced during training, it will give a higher accuracy on the test set where these shortcuts are available.

We show our results on four different configurations of $\beta_1$ and $\beta_2$ values in Table 9. For simplicity, we introduce a new term $L = \beta_1 - \beta_2$, where is $L$ is always greater than 0. We see that for all values of $L$, the accuracy of the test set with shortcuts is greater than the test set without shortcuts, which confirms the presence of an $A \rightarrow T$ bias.

As $L$ increases, the $A \rightarrow T$ model bias increases (as the number of shortcuts increases with $L$). Only $DPA$ reports increasing $A \rightarrow T$ scores as $L$ increases. All other metrics fail to report increasing bias amplification scores. This shows that only $DPA$ correctly captures the amplification of hidden biases (similar to bias amplification from unlabeled objects) in balanced datasets.

## G  Simulating Biases in a Balanced CMNIST Dataset

In this controlled experiment, we use the CMNIST dataset to show that even if attributes (color) and tasks (digit) are balanced in a dataset, models can pick up biases from unlabeled elements and amplify them at test time.

In the CMNIST dataset, we used images of handwritten digits from the original MNIST [29] dataset and added color to them by replacing white pixels with colored pixels. Each image is assigned a single color. The images are the features $X$, digit labels are the task $T$, and color is the protected attribute $A$. Each label (i.e., a digit from 0 - 9) is correlated with a particular color (i.e., 0 - red, 1 - blue, 2 - green and so on). The magnitude of this correlation is controlled using $\alpha$. If $\alpha = 0.7$, $70\%$ image examples of 0 are assigned red, and the remaining $30\%$ images are randomly assigned any of the remaining 9 colors. $\alpha$ allows us to control the bias in the dataset.

Table 10: CMNIST controlled experiment.

|  | $\beta = 0.10$ | $\beta = 0.20$ | $\beta = 0.30$ | $\beta = 0.40$ |
|---|---|---|---|---|
| $DPA_{A \rightarrow T}$ | $0.063 \pm 0.006$ | $0.085 \pm 0.006$ | $0.098 \pm 0.008$ | $0.119 \pm 0.006$ |
| $BA_{A \rightarrow T}$ | $0.000 \pm 0.000$ | $0.000 \pm 0.000$ | $0.000 \pm 0.000$ | $0.000 \pm 0.000$ |
| $LA$ | $0.043 \pm 0.009$ | $0.060 \pm 0.014$ | $0.075 \pm 0.015$ | $0.071 \pm 0.017$ |

We trained a simple CNN model (with depth = 2) on this dataset. We used the trained model to get predictions, and post-processed the predictions using a parameter $\beta$. $\beta$ defines the percentage of predicted labels that are overwritten based on the color in the input image. If $\beta = 0.1$, we randomly select $10\%$ of the predicted labels. These labels are replaced based on the color in the image. This allows us to increase the $A \rightarrow T$ bias in the model predictions. For a fixed value of $\alpha$, $\beta$ allows us to control the model bias. We used accuracy as our $Q$ function.

We started with a balanced CMNIST dataset ($\alpha = 0.1$). We then varied the $\beta$ parameter from $0.1$ to $0.4$ to gradually increase the model bias. We showed the bias amplification captured by different metrics in Table 10.

As we can see in Table 10, $BA_{\rightarrow}$ shows a constant value for varying values of $\beta$. Meanwhile, $LA$ is unable to show a clear trend due to the large confidence intervals. Only $DPA$ shows a monotonic increase in reported bias amplification as $\beta$ value increases. This indicates that only $DPA$ is able to accurately capture the bias amplification in our controlled experiment.

# H   COCO Masking experiment on VGG-16

Table 11: $T \rightarrow A$ bias amplification as the person is progressively masked in the unbalanced and balanced COCO sub-datasets. Attribution scores indicate the $T \rightarrow A$ model bias in the VGG16 model. For the unbalanced sub-dataset, $DPA$ and $BA_{\rightarrow}$ correctly capture the increasing $T \rightarrow A$ model bias. For the balanced sub-dataset, only $DPA$ correctly captures the increasing $T \rightarrow A$ model bias. Subscript shows confidence intervals.

| Dataset Split | Metric | Original | Partial Masked | Segment Masked | Bounding-Box Masked |
|---|---|---|---|---|---|
| | Image | | | | |
| | Attribution Map | | | | |
| Unbalanced | Attribution | $0.6202_{0.0026}$ | $0.6777_{0.0027}$ | $0.7321_{0.0020}$ | $0.7973_{0.0020}$ |
| | $DPA$ (ours) | $0.0034_{0.0006}$ | $0.0195_{0.0007}$ | $0.0214_{0.0010}$ | $0.0242_{0.0004}$ |
| | $BA_{\rightarrow}$ | $0.0029_{0.0002}$ | $0.0072_{0.0005}$ | $0.0108_{0.0007}$ | $0.0140_{0.0007}$ |
| | $Multi_{\rightarrow}$ | $0.0057_{0.0003}$ | $0.0091_{0.0005}$ | $0.0109_{0.0005}$ | $0.0219_{0.0011}$ |
| | $LA$ | $0.0005_{0.0000}$ | $0.0068_{0.0009}$ | $0.0032_{0.0004}$ | $0.0011_{0.0012}$ |
| Balanced | Attribution | $0.6292_{0.0027}$ | $0.6992_{0.0024}$ | $0.7367_{0.0019}$ | $0.8065_{0.0183}$ |
| | $DPA$ (ours) | $0.0019_{0.0002}$ | $0.0024_{0.0004}$ | $0.0068_{0.0009}$ | $0.0100_{0.0011}$ |
| | $BA_{\rightarrow}$ | $0.0000_{0.0000}$ | $0.0000_{0.0000}$ | $0.0000_{0.0000}$ | $0.0000_{0.0000}$ |
| | $Multi_{\rightarrow}$ | $0.0035_{0.0002}$ | $0.0056_{0.0004}$ | $0.0060_{0.0003}$ | $0.0099_{0.0010}$ |
| | $LA$ | $0.0012_{0.0002}$ | $0.0055_{0.0016}$ | $0.0026_{0.0005}$ | $0.0065_{0.0011}$ |

We recreated our COCO masking experiment from section 4.1 with a VGG16 [16] model. The results are shown in Table 11.

For the unbalanced COCO sub-dataset (row 2 of Table 11), $DPA$ and $BA_{\rightarrow}$ report increasing $T \rightarrow A$ scores across versions, correctly capturing the growing model bias (indicated by the growing model attribution scores). While $Multi_{\rightarrow}$ shows the same trend as these metrics, it is not a reliable metric as it cannot capture negative bias amplification (discussed in Section E).

For the balanced sub-dataset, we observe growing model attribution scores indicating a rising $T \rightarrow A$ model bias. $DPA$ correctly captures this trend by reporting increasing $T \rightarrow A$ scores. While $Multi_{\rightarrow}$ shows the same trend as $DPA$, as discussed earlier, $Multi_{\rightarrow}$ is not a reliable metric. $LA$ fails to capture the growing model bias as it is not directional. $BA_{\rightarrow}$ assumes that a balanced dataset is unbiased. Hence, it fails to capture the growing model bias and constantly reports zero bias amplification.

We also show additional examples for attribution maps for the VGG16 model on images from our balanced COCO sub-dataset in Figure 5. We observe in these images that as we increase masking on

the person, i.e., attribution examples from left to right, the attribution shifts from the person to either the background or unlabeled task objects. This shows that in balanced datasets, as masking increases, the model relies more on unlabeled background objects to determine the gender of the person.

| Original Image | No Masking | Partial Masking | Segment Masking | Box Masking |

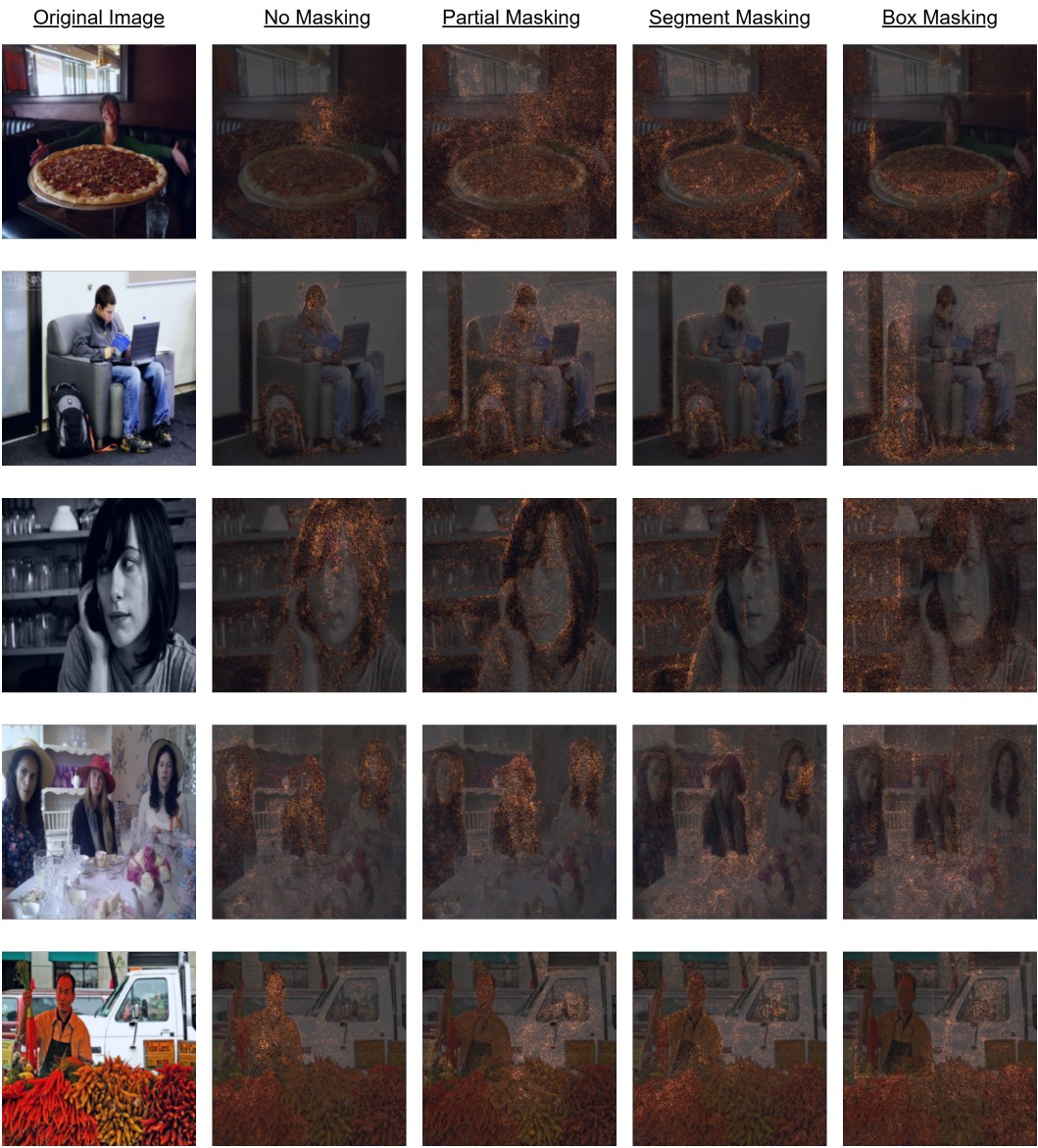

Figure 5: VGG-16 model attributions for images from the balanced COCO sub-dataset. For all cases, as the person in the image is progressively masked, the VGG-16 often relies on unlabeled background objects to predict gender.

# I  ImSitu Experiment: $T \rightarrow A$ results

In Table 12, we showed the sensitivity and bias amplification scores in the $T \rightarrow A$ direction for the unbalanced sub-dataset. In Table 13, we showed corresponding results for the balanced sub-dataset. For both tables, we ranked all models (from 1 to 9) by their $Sen_{T \rightarrow A}$ scores, from least biased to most biased. We checked which bias amplification metrics produced rankings closest to this model bias ranking. This allowed us to see which metrics best captured changes in model bias.

In the unbalanced case (Table 12), $DPA$'s ranking is closest to the model bias ranking — it differs by only three positions. All other metrics show significant mismatches.

In the balanced case (Table 13), $DPA$'s ranking is closest to the model bias ranking — it again differs by only three positions. Other metrics, like $LA$ and $Multi_\rightarrow$, differ significantly. Metrics like $BA_\rightarrow$ and $BA_{MALS}$ report zero bias amplification because they assume that a balanced dataset has no bias.

Table 12: $T \rightarrow A$ results for the ImSitu unbalanced sub-dataset. Models are shown in increasing order of model bias, based on their $Sen_{T \rightarrow A}$ scores. For each metric, the number in parentheses shows the rank it assigned to that model — where rank 1 means the lowest (most negative) bias amplification and rank 9 means the highest (most positive). Red numbers highlight where the metric's rank differs from the model bias rank assigned by $Sen_{T \rightarrow A}$. $DPA$ is the most reliable metric here, as its model rankings closely match the $Sen_{T \rightarrow A}$ rankings. All values are scaled by 100.

| Models | $Sen_{T \rightarrow A}$ | $DPA_{T \rightarrow A}$ | LA | $BA_{T \rightarrow A}$ | $Multi_{T \rightarrow A}$ | $BA_{MALS}$ |
|---|---|---|---|---|---|---|
| SqueezeNet [21] | 0.000 (1) | −0.013 (3) | 0.009 (1) | −18.26 (1) | 18.90 (9) | 25.59 (9) |
| ViT_b_32 [15] | 0.033 (2) | −0.025 (1) | 0.209 (6) | −6.891 (6) | 7.252 (5) | 0.248 (8) |
| Wide ResNet50 [22] | 0.036 (3) | −0.020 (2) | 0.253 (8) | −4.494 (7) | 5.289 (3) | 0.061 (4) |
| Wide ResNet101 [22] | 0.098 (4) | −0.012 (4) | 0.281 (9) | −3.612 (8) | 4.790 (2) | 0.109 (3) |
| MaxViT [20] | 0.188 (5) | 0.000 (5) | 0.130 (4) | −8.143 (4) | 17.61 (8) | 0.212 (6) |
| ViT_b_16 [15] | 0.190 (6) | 0.001 (6) | 0.195 (5) | −7.214 (5) | 7.719 (4) | 0.229 (7) |
| MobileNet V2 [23] | 0.566 (7) | 0.009 (7) | 0.087 (3) | −9.085 (3) | 9.761 (7) | 0.193 (5) |
| Swin Tiny [24] | 2.800 (8) | 0.012 (8) | 0.064 (2) | −9.782 (2) | 10.86 (6) | −0.082 (1) |
| VGG16 [16] | 2.938 (9) | 0.013 (9) | 0.210 (7) | −1.620 (9) | 3.979 (1) | −0.045 (2) |

Table 13: $T \rightarrow A$ results for the ImSitu balanced sub-dataset. This table follows the same setup as Table 12. Red numbers highlight where the metric's rank differs from the model bias rank assigned by $Sen_{T \rightarrow A}$. $DPA$ is the most reliable metric here, as its model rankings exactly match the model bias ranking given by $Sen_{T \rightarrow A}$. All values are scaled by 100.

| Models | $Sen_{T \rightarrow A}$ | $DPA_{T \rightarrow A}$ | LA | $BA_{T \rightarrow A}$ | $Multi_{T \rightarrow A}$ | $BA_{MALS}$ |
|---|---|---|---|---|---|---|
| SqueezeNet [21] | 0.000 (1) | 0.019 (3) | 0.003 (8) | 0.000 (1) | 0.208 (2) | 0.000 (1) |
| Wide ResNet50 [22] | 0.107 (2) | 0.016 (1) | 0.002 (7) | 0.000 (1) | 3.580 (8) | 0.000 (1) |
| Wide ResNet101 [22] | 0.122 (3) | 0.018 (2) | 0.000 (3) | 0.000 (1) | 1.815 (3) | 0.000 (1) |
| MaxViT [20] | 0.261 (4) | 0.021 (4) | −0.002 (1) | 0.000 (1) | 0.022 (1) | 0.000 (1) |
| MobileNet V2 [23] | 0.297 (5) | 0.021 (5) | 0.007 (9) | 0.000 (1) | 1.975 (4) | 0.000 (1) |
| ViT_b_16 [15] | 0.310 (6) | 0.021 (6) | 0.000 (2) | 0.000 (1) | 2.174 (5) | 0.000 (1) |
| Swin Tiny [24] | 0.345 (7) | 0.022 (7) | 0.001 (5) | 0.000 (1) | 2.628 (7) | 0.000 (1) |
| VGG16 [16] | 0.497 (9) | 0.047 (8) | 0.000 (3) | 0.000 (1) | 5.638 (9) | 0.000 (1) |
| ViT_b_32 [15] | 0.626 (9) | 0.174 (6) | 0.001 (5) | 0.000 (1) | 2.584 (6) | 0.000 (1) |

## J  Additional Experiment Details

Given below are additional details about the hyperparameters used for $DPA$ and $LA$ metrics in different experiments mentioned in the paper. Since $BA_{MALS}$, $BA_\rightarrow$, and $Multi_\rightarrow$ do not train an attacker function, no such hyperparameters are required for them. For all experiments, we used (1/ cross-entropy) as our quality function ($Q$).

Table 14: COCO Masking additional parameters

| Parameter | Optimizer | Attacker Depth | Learning Rate | Num. epochs | Batch size |
|---|---|---|---|---|---|
| $DPA$ | Adam | 2 | 0.001 | 100 | 64 |
| $LA$ | Adam | 2 | 0.001 | 100 | 64 |

Table 15: ImSitu additional parameters

| Parameter | Optimizer | Attacker Depth | Learning Rate | Num. epochs | Batch size |
|---|---|---|---|---|---|
| $DPA$ | Adam | 2 | 0.001 | 100 | 128 |
| $LA$ | Adam | 2 | 0.001 | 100 | 128 |

Table 16: COMPAS additional parameters

| Parameter | Optimizer | Attacker Depth | Learning Rate | Num. epochs | Batch size |
|---|---|---|---|---|---|
| $DPA$ | Adam | 2 | 0.005 | 50 | 512 |
| $LA$ | Adam | 2 | 0.005 | 50 | 512 |

## K  Behavior of different directional metrics

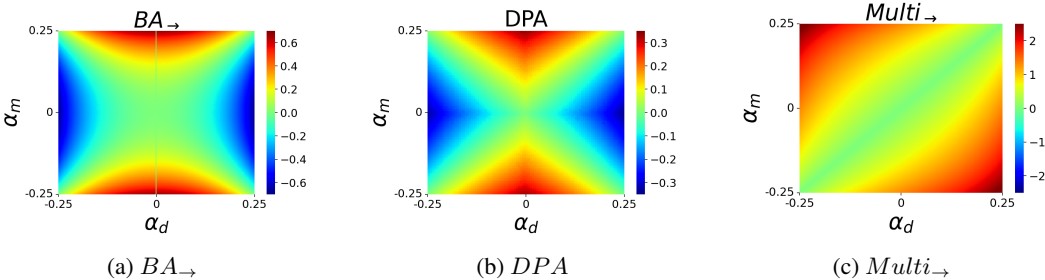

(a) $BA_\rightarrow$          (b) $DPA$          (c) $Multi_\rightarrow$

Figure 6: Bias amplification heatmap for different configurations of the dataset (X-axis) and model predictions (Y-axis). $\alpha_d$ creates different configurations of the dataset, while $\alpha_m$ creates different configurations of the model predictions. $BA_\rightarrow$ and $DPA$ show similar behavior (except when the dataset is balanced). However, $Multi_\rightarrow$ always reports positive bias amplification.

To analyze the behavior of directional metrics, we simulated a dataset with a protected attribute $A$ ($A = 0, 1$) and task $T$ ($T = 0, 1$). Initially, each $A$ and $T$ pair had an equal probability of 0.25. To introduce bias in the dataset, we modified the probabilities by adding a term $\alpha_d$ to the group $\{A = 0, T = 0\}$ and subtracting $\alpha_d$ from $\{A = 1, T = 1\}$. Here, $\alpha_d$ ranges from $-0.25$ to $0.25$ in steps of $0.005$. The dataset is balanced only when $\alpha = 0$ but becomes increasingly unbalanced as $\alpha_d$ deviates from 0. In the same manner, we use $\alpha_m$ to introduce bias in the model predictions.

With $\alpha_d$ and $\alpha_m$ ranging from $-0.25$ to $0.25$, we create 100 different versions of the dataset and model predictions, respectively. For each metric, we plot a $100 \times 100$ heatmap of the reported bias amplification scores. Each pixel in the heatmap represents the bias amplification score for a specific {dataset, model} pair.

Figure 6 shows the heatmaps for all metrics. Figures 6a and 6b display the bias amplification heatmaps for $BA_\rightarrow$ and $DPA$, respectively, with similar patterns. However, $BA_\rightarrow$ (Figure 6a) shows a distinct vertical green line in the center, indicating that when the dataset is balanced ($\alpha_d = 0$ on the X-axis), bias amplification remains at 0, regardless of changes in model's bias (varying $\alpha_m$ values on the Y-axis). In contrast, $DPA$ (Figure 6b) accurately detects non-zero bias amplification whenever there is a shift in bias in either the dataset or model predictions, making it a more reliable metric. $Multi_\rightarrow$ (as shown in Figure 6c) reports positive bias amplification in all scenarios, making it unreliable.

## L  Runtime Comparison

In Table 17, we compared the time taken by different metrics to compute bias amplification on the COCO dataset. This computation was performed over an Intel Core i7 165H processor without any GPU acceleration. Note that bias (or bias amplification) analysis is usually conducted offline (and not in real time). While $DPA$ is several magnitudes more time-consuming than the other methods, an execution time of two minutes is acceptable for most offline applications.

Table 17: Runtime comparison for different bias amplification metrics on the COCO dataset.

|           | $BA_{MALS}$ | $BA_\rightarrow$ | $Mutli_\rightarrow$ | $LA$ | $DPA$ |
|-----------|-------------|------------------|---------------------|------|-------|
| Time (ms) | 57.57       | 150.16           | 453.11              | $1.90 \times 10^5$ | $1.20 \times 10^5$ |

