# OpenReview forum: "DPA: A one-stop metric to measure bias amplification in classification datasets"
_NeurIPS.cc/2025/Conference — NeurIPS 2025 poster_

### Official Review · Reviewer_dTPG · 2025-06-29

**Clarity:** 3
**Significance:** 3
**Originality:** 4
**Rating:** 4
**Confidence:** 3

**Summary:**

The authors propose Directional Predictability Amplification (DPA), a new metric for measuring bias amplification in machine learning models. Unlike previous metrics, DPA captures the directionality of bias amplification—even in balanced datasets—and accounts for both its positive and negative amplification. The authors demonstrate DPA's effectiveness on both tabular and image datasets.

**Questions:**

1. Is there a toy example (e.g., Colored MNIST) where model bias amplification arises due to easily learned shortcut features used as attributes, which can effectively demonstrate the usefulness of the proposed metric?


2. How can the benefits of the new metric be utilized in real-world scenarios? Are there tasks where DPA offers improvements that other metrics cannot?

**Ethical Concerns:**

["NO or VERY MINOR ethics concerns only"]

**Final Justification:**

I found the Colored MNIST experiment in the rebuttal particularly compelling.

By evaluating DPA in a controlled setting with limited confounding factors, the authors effectively demonstrated its advantage over existing metrics.

The paper identifies key limitations of prior approaches and presents a well-motivated alternative.

While the contribution feels more incremental than entirely novel, I believe it is a meaningful step forward.

Therefore, I would lean toward a borderline accept.

**Limitations:**

yes

**Paper Formatting Concerns:**

None.

**Quality:**

3

**Strengths And Weaknesses:**

Strengths:
1. Systematic analysis of the limitations of previous metrics, both conceptually and experimentally.
2. Thorough explanation of the results for each experiment was good.
3. Diverse experiments on the architectural selection of attacker models.

Weaknesses:
1. Requires training additional models for measurement.
2. The authors use attribution score measures and sensitivity ranking to show the tendency of model bias amplification. However, these are proxies for model bias, making it difficult to compare the absolute effectiveness of metrics (i.e., they only show whether the metric aligns with the tendency or not).

---

> ### Author Rebuttal · Authors · 2025-07-30
>
> We thank the reviewer for their constructive feedback. We address each point below.
>
> $\newline$
>
> **Question 1**:
>
> We have shown a controlled experiment on COCO using OPS[1] based shortcuts in Appendix F. But based on your recommendation, we have also added another experiment on the CMNIST dataset to get more empirical proof.
>
> Experiment Setup:
>
> The experiment is controlled by two control variables $\alpha$ and $\beta$ to control the dataset and model bias, respectively.
>
> In the CMNIST dataset, we modify the traditional MNIST dataset by replacing the black and white images with their colored versions.  The images are the features ($X$), digit labels are the task ($T$), and color is the protected attribute ($A$). Each label (i.e., digit from 0-9) is correlated with a particular color (i.e., 0- red, 1- blue, 2- green, …). The magnitude of this correlation is controlled using alpha. I.e., if $\alpha =0.7$, 70% of examples of 0 are red, and the remaining 30% are randomly assigned any of the remaining 9 colors. Thus, $\alpha$ allows us to control the bias in the dataset.
>
> A simple CNN model (with depth = 2) is trained on this dataset. The trained model is used to get predictions. The predictions go through post-processing based on $\beta$. $\beta$ defines the percentage of predicted labels that are overwritten based on the color in the input image. I.e., if $\beta=0.1$, 10% of the predicted labels will be randomly selected. These labels will then be replaced based on the color in the image. This allows us to increase the $A\rightarrow T$ bias in the model predictions. Thus, for a fixed value of $\alpha$, $\beta$ allows us to control the model bias. For $Q$ function, accuracy is used.
>
> Results:
>
> $\newline$
>
> $\beta = 0.0$
>   |                   | $\alpha = 0.60$     | $\alpha = 0.65$     | $\alpha = 0.70$     | $\alpha = 0.75$     |
> |-------------------|---------------------|---------------------|---------------------|---------------------|
> | $\Psi_{A}^{D}$  | $0.5816 \pm 0.010$   |    $0.6536 \pm 0.008$ | $0.6944 \pm 0.003$ | $0.7507 \pm 0.012$ |
>
> As we see from the table above, $\Psi_{A}^{D}$ shows a clear increasing trend as the controlled variable $\alpha$ increases. I.e., the dataset bias detected by $DPA$ increases directly as the actual dataset bias (controlled by $\alpha$) increases. This controlled experiment allows us to verify that $DPA$ is capable of measuring the dataset bias.
>
> $\newline$
>
> $\alpha = 0.75$
> |                         | $\beta = 0.00$       | $\beta = 0.05$       | $\beta = 0.10$       | $\beta = 0.15$       |
> |-------------------------|----------------------|----------------------|----------------------|----------------------|
> | $\text{DPA}_{A \rightarrow T}$   | $0.021 \pm 0.008$   |    $ 0.057 \pm 0.007$ | $0.094 \pm 0.017$ | $0.132 \pm 0.016$ |
>
> As we see from the table above, $\text{DPA}_{A\rightarrow T}$ shows a clear increasing trend as the controlled variable $\beta$ increases. I.e., the bias amplification detected by $DPA$ increases directly as the actual model bias (controlled by $\beta$) increases. This controlled experiment allows us to verify that $DPA$ is capable of accurately measuring bias amplification.
>
>
> We shall be happy to add this experiment in a more detailed fashion in the camera-ready version.
>
> $\newline$
>
> **Question 2**:
>
> Directional bias amplification metrics (like $BA_{\rightarrow}$ and DPA) can be used to verify if bias mitigation strategies work correctly. Let us revisit an example shown by Wang et al. [2] where the Equalizer model, a model that mitigates gender bias amplification in image captions, accidentally increases $A\rightarrow T$ bias amplification.
>
> Consider an image of a woman holding a book, and the following gender-biased caption: “A man holding a book”. The Equalizer model corrects the gender of this caption: “A woman holding a kite”. However, it incorrectly changes the object in the caption. While the Equalizer model mitigates $T\rightarrow A$, it accidentally increases $A\rightarrow T$ bias amplification.
>
> For this caption, $DPA$ becomes useful if $A$ and $T$ are balanced in the training dataset (ML practitioners often balance datasets, assuming that balancing gives them an unbiased dataset). If $A$ and $T$ are balanced, even though the Equalizer model predicted the object incorrectly (kite instead of book) while correcting gender, the $A\rightarrow T$ bias amplification reported by $BA_{\rightarrow}$ would be 0. This is because $BA_{\rightarrow}$ incorrectly assumes that a model cannot amplify biases if it is trained on a balanced dataset. However, DPA would report the intended positive bias amplification in $A\rightarrow T$.
>
> $\newline$
>
> References
>
> [1] Shutong Wu, , Sizhe Chen, Cihang Xie, Xiaolin Huang. "One-Pixel Shortcut: On the Learning Preference of Deep Neural Networks." The Eleventh International Conference on Learning Representations . 2023.
>
> [2] Angelina Wang and Olga Russakovsky. Directional bias amplification. In International Conference on Machine Learning, pages 10882–10893. PMLR, 2021.

---

> > ### Comment · Reviewer_dTPG · 2025-08-02
> > **Follow-up Comment on Question 1**
> >
> > Thank you for conducting the experiments on the more intuitive CMNIST dataset — it significantly improves clarity. Given that the dataset has few confounding factors and is syntactic in nature, it offers a more controlled setting for demonstrating the method’s effectiveness.
> >
> > Would it be possible to demonstrate that directionality is not captured by LA on the same dataset? It would also strengthen the work if you could show that DPA clearly outperforms the other methods compared in this paper, such as BA→, LA.

---

> ### Author Response · Authors · 2025-08-04
> **Addition of other metrics**
>
> Thank you for your recommendation! It is definitely possible to show the superiority of our metric using the above experiment.
>
> To show the efficacy of our metric, we started with a balanced CMNIST dataset (i.e., $\alpha$ = 0.1). We then varied the $\beta$ parameter from 0.1 to 0.4 to gradually increase the model bias. We showed the bias amplification captured by different metrics in the table below.
>
> |                         | $\beta = 0.10$       | $\beta = 0.20$       | $\beta = 0.30$       | $\beta = 0.40$       |
> |-------------------------|----------------------|----------------------|----------------------|----------------------|
> | $\text{DPA}_{A \rightarrow T}$ | $0.063 \pm 0.006$ | $0.085 \pm 0.006$ | $0.098 \pm 0.008$ | $0.119 \pm 0.006$ |
> | $BA_{A\rightarrow T}$ | $0.000 \pm 0.0000$ | $0.000 \pm 0.0000$ | $0.000 \pm 0.0000$ | $0.000 \pm 0.0000$ |
> | $\text{LA}$ | $0.043 \pm 0.009$ | $0.060 \pm 0.014$ | $0.075 \pm 0.015$ | $0.071 \pm 0.017$ |
>
> Since the dataset is balanced, $BA_{\rightarrow}$ reported zero bias amplification for all $\beta$ values. LA was unable to capture the increasing trend as it is not a directional metric. Only DPA correctly captured the increasing trend. This shows that only DPA can capture directionality in a balanced dataset.
>
> We will add a more detailed version of this experiment in our supplementary for the camera-ready version.
>
> We hope this resolves your queries. If there is anything else we can do to encourage you to increase your score, please let us know!

---

> > ### Comment · Reviewer_dTPG · 2025-08-06
> > **Response to Rebuttal**
> >
> > Thank you for conducting the additional experiments. The results make the benefit of DPA clearer, and I appreciate the effort.
> > Based on this, I am happy to raise my score to a 4.
> > If this analysis is included in the final version of the paper, I believe it will further help readers understand the effectiveness of DPA.

---

### Official Review · Reviewer_SYXG · 2025-06-30

**Clarity:** 3
**Significance:** 3
**Originality:** 3
**Rating:** 4
**Confidence:** 4

**Summary:**

This paper introduces Directional Predictability Amplification (DPA), a novel metric for measuring bias amplification in classification datasets. The authors address limitations in existing co-occurrence-based metrics (BA→, Multi→, BAMALS) which fail to detect bias amplification in balanced datasets or cannot identify negative bias amplification. DPA uses predictability-based measurements with attacker functions to quantify how well protected attributes (e.g., gender) can predict task labels, providing directional analysis whilst working across both balanced and unbalanced datasets. The metric reports bounded scores in [-1, 1] and measures relative rather than absolute changes in predictability. Experiments on COMPAS, COCO, and ImSitu datasets demonstrate DPA's superiority over existing metrics.

**Questions:**

1. Attacker Function Sensitivity: Line 79 claims DPA is "minimally sensitive" to attacker function choice. Could you provide more rigorous theoretical or empirical analysis of this claim? Under what conditions might this sensitivity become problematic?
2. Generalisation Beyond Gender: Your experiments focus predominantly on gender bias. How does DPA perform with other protected attributes like race, age, or socioeconomic status? Could you demonstrate the metric's effectiveness on non-visual domains (text, audio)?
3. Computational Complexity: What are the computational requirements of DPA compared to co-occurrence-based metrics? How does training time for attacker functions scale with dataset size and complexity?
4. Failure Modes: Under what theoretical or practical conditions might DPA produce misleading results? How can practitioners identify when the metric may be unreliable?
5. Individual A-T Pair Analysis: You acknowledge in limitations that DPA cannot measure bias amplification for specific A-T pairs. Could you elaborate on potential approaches to address this limitation, perhaps through ensemble methods or hierarchical analysis?

**Ethical Concerns:**

["NO or VERY MINOR ethics concerns only"]

**Final Justification:**

The rebuttal addressed several practical concerns effectively. The authors provided useful timing data showing DPA requires ~2 minutes for COCO dataset analysis, which is acceptable for offline bias evaluation. They clarified their scope includes racial bias evaluation (COMPAS dataset) beyond gender-focused experiments and acknowledged key failure modes like underfitting/overfitting with proposed monitoring approaches. These responses demonstrate good engagement and practical awareness.
However, critical limitations remain unresolved that prevent a higher rating. Most significantly, the core claim that DPA is "minimally sensitive" to attacker function choice lacks sufficient empirical validation - the evidence in Appendix D is limited, and authors acknowledge time constraints prevented deeper analysis. The evaluation scope remains narrow (predominantly visual domains, gender bias) despite claims of broad applicability for a "one-stop" metric. Additionally, the work lacks theoretical analysis of when DPA might fail beyond basic scenarios. I assign highest weight to the technical soundness and genuine contribution of addressing bias amplification in balanced datasets, but the unresolved robustness validation significantly undermines confidence in the metric's reliability across diverse applications. The work merits publication as it addresses important gaps in fairness research, but requires stronger validation and broader evaluation to achieve its ambitious goals as a comprehensive bias amplification solution.

**Limitations:**

YEs

**Paper Formatting Concerns:**

No major formatting issues observed.

**Quality:**

3

**Strengths And Weaknesses:**

## Strengths
- Quality: The paper presents a technically sound solution to genuine limitations in existing bias amplification metrics. The mathematical formulation is rigorous, with proper normalisation ensuring bounded outputs and relative change measurement accounting for dataset bias. The experimental methodology is comprehensive, using multiple datasets (tabular and image) with proper controls including progressive masking experiments and synthetic data validation.
- Clarity: The paper is well-structured with clear mathematical notation and effective use of tables and figures. The motivation is well-established through concrete examples (e.g., ImSitu cooking bias), and the comparison with existing metrics is systematic and thorough.
- Significance: This work addresses an important problem in fairness research. The ability to detect bias amplification in balanced datasets is particularly valuable, as practitioners often assume balanced datasets are unbiased. The directional analysis capability provides actionable insights for targeted bias mitigation.
- Originality: The combination of predictability-based measurement with directionality is novel. Whilst building on Wang et al.'s leakage amplification concept, the addition of directional analysis and the relative change formulation represents a meaningful advancement.

Weaknesses
- Limited Scope: The evaluation focuses heavily on gender bias across image datasets. Testing on other protected attributes (race, age) and domains (text, audio) would strengthen generalisability claims. The choice of attacker function sensitivity claims require more rigorous validation.
- Theoretical Gaps: The paper lacks theoretical analysis of when DPA might fail or produce misleading results. The relationship between attacker function choice and metric reliability needs deeper investigation beyond the limited robustness experiments.
- Practical Considerations: Limited discussion of computational requirements compared to simpler co-occurrence metrics. The need to train attacker functions adds complexity that may hinder adoption.

---

> ### Author Rebuttal · Authors · 2025-07-30
>
> We thank the reviewer for their insightful feedback. We address their questions below.
>
> $\newline$
>
> **Question 1**:
>
> Given the time constraints, we were not able to provide any further analysis on the attacker function apart from the one currently present in Appendix D. If you recommend, we could add an experiment showing the reduced sensitivity by varying other model parameters (e.g., architecture, depth, and activation functions) by the camera-ready deadline.
>
> High attacker sensitivity can be especially problematic as it can cause misleading results. High attacker sensitivity can lead to different bias amplification scores for the same model or change the relative magnitude of biases across different models. I.e., for bias amplification in image captioning, Bakr et. al[1] show that the LIC[2] metric is inconsistent in its ranking of bias amplification in different models due to attacker model sensitivity.
>
>
> $\newline$
>
> **Question 2**:
>
> To stay consistent with previous works on bias amplification, we measured bias amplification in visual and synthetic datasets on attributes like gender and race. Our experiment on racial biases using the COMPAS dataset (COMPAS is a synthetic dataset) is available in the Appendix – Section E.
>
> Text or NLP-focused datasets (e.g., image captioning datasets) have semantic nuances. For such datasets, there are specialized metrics (like LIC) that incorporate text-based semantics when computing bias amplification. The metrics discussed in this work (and related previous works) are largely focused on classification problems in the visual and synthetic domains. We did not investigate audio datasets as we were not aware of publicly available datasets with protected attribute information. It would be helpful if you could suggest audio datasets that could be studied for bias amplification.
>
> $\newline$
>
> **Question 3**:
>
> In the table below, we compared the time taken by different metrics to compute bias amplification on the COCO dataset. This computation was performed over an Intel Core Ultra 7 165H processor without any GPU acceleration.
>
> |           | $BA_{MALS}$    | $BA_{\rightarrow}$        | $Multi_{\rightarrow}$       | $LA$               | $DPA$                 |
> |-----------|-------------|------------|------------|-----------------------|----------------------|
> | Time (ms) | 57.57 ms    | 150.16 ms  | 453.11 ms  | $1.90 \times 10^{5}$ ms | $1.2 \times 10^{5}$ ms |
>
> Note that bias amplification is often measured in offline analysis. Hence, execution time is not the utmost priority. Moreover, DPA takes ~ 2 minutes on the COCO dataset, which is sufficient for an offline analysis in most applications.
>
> With regards to training time scaling with dataset size, we expect a linear trend. The training process itself is similar to conventional model training and hence, would follow the same linear trend with respect to increasing dataset size.
>
> $\newline$
>
> **Question 4**:
>
> Poor choice of attacker models can lead to misleading or unreliable results. If the attacker model grossly underfits or overfits the training data, it can lead to bias amplification being over- or under-reported. Monitoring the training and test loss of the attacker models can help identify such misleading values. We have briefly talked about this in our Limitations section. If you suggest that these details would be beneficial for readers, we can add these details in the Limitations section by the camera-ready version.
>
> $\newline$
>
>  **Question 5**:
>
> There are several potential approaches to analyze individual {A, T} pairs. One approach can be to use feature attribution. Assume we want to measure dataset and model bias in $ \ A\rightarrow T$. If we one-hot encode both $A$ and $T$, we can use feature attribution to find out the influence of a specific attribute $A=A_{i}$, in predicting a specific task $T=T_k$. This attribution value can be used as a weak proxy to measure dataset or model bias for this ${A_i, T_k}$ pair. Causal attacker models or interpretable attackers are other avenues for a similar ${A, T}$ pair analysis. We can add a section discussing such ideas for future work.
>
> $\newline$
>
> References
>
> [1] Eslam Mohamed Bakr, , Pengzhan Sun, Li Erran Li, Mohamed Elhoseiny. "ImageCaptioner$^2$: Image Captioner for Image Captioning Bias Amplification Assessment." (2023).
>
> [2] Hirota, Yusuke, Yuta, Nakashima, Noa, Garcia. "Quantifying Societal Bias Amplification in Image Captioning." 2022 IEEE/CVF Conference on Computer Vision and Pattern Recognition (CVPR). 2022.

---

> > ### Author Response · Authors · 2025-08-07
> >
> > Thank you for taking the time to review our paper and share your feedback. Do you have any additional questions or concerns related to your rating that we have not addressed?

---

> > ### Comment · Reviewer_SYXG · 2025-08-08
> >
> > Thank you for your detailed responses. The rebuttal provides helpful clarifications and demonstrates good awareness of the work's limitations. Your acknowledgment of time constraints regarding attacker sensitivity analysis is understandable, and the proposed additional experiments would strengthen the work. The timing comparison for computational complexity shows reasonable performance for offline analysis (~2 minutes for COCO), and your identification of underfitting/overfitting as key failure modes is appropriate. The feature attribution approach for individual A-T pair analysis represents a promising future direction.
> >
> > However, the rebuttal highlights limitations rather than fully resolving them. The narrow evaluation scope (primarily visual domains, gender bias) remains, despite the COMPAS racial bias work. The current evidence for attacker function robustness claims remains limited to specific architectures and datasets. While your distinction between visual classification metrics and text-based metrics like LIC is reasonable, it does constrain generalisability claims.
> >
> > I maintain my original rating of 4: Borderline Accept.

---

### Official Review · Reviewer_n6n6 · 2025-07-02

**Clarity:** 2
**Significance:** 4
**Originality:** 3
**Rating:** 4
**Confidence:** 1

**Summary:**

### Problem:
_a.k.a. What problem/question does this work purport to solve/answer?_

A particular aspect of bias within ML/AI that is particularly concerning is bias amplification -- the phenomena in which a ML model does not (as naive theory might suggest) automatically learn to ameliorate data biases but actually amplify them in downstream use. In order to effectively reduce bias amplification, we need to be able to measure it. This paper proposes a new method to do so that solves several key challenges:
  1. Directionality
  2. Working across balanced and unbalanced datasets
  3. Works with both positive and negative bias

### Impact:
_a.k.a. Is the problem/question impactful?_

Bias amplification is a critical property that intersects both with the high-impact nature of fairness challenges and the possibility of meaningful technical solution through new innovations, so I feel this is a key direction for impact.

### Proposed Solution:
_a.k.a. How does this work claim it will solve/answer the problem/question?_

They define their metric to be the relative change in predictability of a protected attribute from a real vs. predicted label (or the reverse).

### Novelty/Why hasn't this been done before:
_a.k.a. How does this work differ from the existing literature?_

Prior methods have examined an absolute change in predictability, rather than a relative change. The authors claim this is a significant problem, because it conflates a small relative increase in bias in settings with high dataset bias and a large increase in relative bias for settings with low dataset bias.  As far as I can tell, this difference is the largest source of novelty in the work.

### Evidence that this work solves the problem:
_a.k.a. How does this work prove that their solution/answer is valid?_

The authors employ a very clever experiment (inspired in part by some prior work, I believe) in which they artificially introduce a setting in which the model must amplify bias in order to make effective predictions at all, then sees to what extent their metric catches that bias amplification versus competing approaches.

**Questions:**

1. I don't understand why you flip some of the predicted labels T based on M's accuracy. Can you further explain this? What is the error case you're concerned with here?

2. How would a technique like multicalibration impact the success of your approach here? I think that warrants testing.

**Ethical Concerns:**

["NO or VERY MINOR ethics concerns only"]

**Final Justification:**

See my rebuttal response and original review for justification.

**Limitations:**

No comment

**Paper Formatting Concerns:**

You need to explicitly define that DPA_* = 0 if the denominator is zero. I understand that that is a reasonable thing to assume, but 0 / 0 is not well defined, so you need to explicitly state that for your claim that it is bounded to be true.

**Quality:**

2

**Strengths And Weaknesses:**

## Strengths
### Key Strengths
_a.k.a. reasons I would advocate this paper be accepted._

  1. This problem is very high impact and new metrics to detect bias amplification are important.
  2. I like your controlled masking experiments as a vehicle to artificially inflate the expected bias amplification.

### Minor Strengths
_a.k.a. things I like but would not be sufficient to motivate acceptance in isolation._

  1. I like the use of proxy signals (like attribution maps and CAVs) to reinforce what the expected behavior is (though I would not trust them absolutely).

## Weaknesses
### Key Weaknesses
_a.k.a. reasons I would advocate this paper be rejected._

#### Flaw in your results
I do not believe your results can be correct as presented -- in particular the relative performance of LA vs. DPA. Let's let $Q$ be $\lambda$ -- both are (as you describe them) measures of how well the attribute can be predicted from the task in the dataset vs. the predicted task by the model, unless I've misunderstood. $Q_D$ in your experiment should be fixed across all masking levels -- in the data, the task predicts the attribute in a manner that is unaffected by masking. $Q_M$ changes as you adjust the masking strategy and rate -- and in particular _should_ if your experiment has internal validity be increasing as we move to the right (in Table 2). This means that LA -- defined as $Q_M$ - $Q_D$ -- _should also be increasing monotonically as we move to the right_. But, instead, you have the last value in the Bounding-Box Masked column being lower than any other. Why is this? What has gone wrong in the use of the LA metric or my understanding of it to deviate from what should be the result here?

Without understanding or explaining this, this result undercuts your entire argument, because it calls the internal validity of your experiment or the validity of the baselines into question, so this is essential to correct.

#### Insufficient Novelty
Your method is functionally just a prior method (LA) divided by the sum of the two quality terms. The fact that DPA can disentangle the two directions is merely because you repeat the analysis of LA but with a swapped internal definition of A and T. This is very much stretching the novelty bar for a venue like NeurIPS. This weakness may be overcome by your results, which are very promising, but it is a major challenge nonetheless. Fundamentally, this weakness compounds some of the skepticism about results, as -- even if your reported improvements over LA are valid -- it is such a tiny change to improve LA in a manner that re-captures your proposed approach.

### Minor Weaknesses
_a.k.a. things I dislike but would not be sufficient to motivate rejection in isolation._

  1. Your evaluation metrics need to be summarized in some way so that we can read off whether DPA is superior to other metrics in one pass, rather than needing to read all the numbers in Table 2. E.g., report the correlation between the % masked and the metric, or something like that.
  2. I'd love for your metric to be more theoretically principled, rather than just a subjective justification on a change from an absolute difference to a relative one. E.g., is there a probabilistic perspective you can leverage to suggest a relative metric is superior under a given biasing method?
  3. You should generate some purely synthetic experiments that unambiguously show the improvement of your method, so you don't need to rely on heuristics or proxy variables to establish what the "right" bias direction should be.

## Final Recommendation:
_My overall recommendation for this paper, and its justification._

Reject

I recommend reject because, without further clarification of the results, I cannot trust the conclusions presented (see key weakness 1). If this is clarified and my misunderstandings addressed, I could see my score rising as high as a Marginal Accept even with the concern regarding novelty.

## What do I want to see changed:
### Changes to Increase my Score:
_What things, were they provided to my satisfaction, would motivate me improve my score?_

Clarify (and convince me) why the LA results do not undermine your argument and experimental coherence and more generally re-present your results or present additional results that establish the superiority of DPA in a more convincing, internally consistent manner. In addition, some further justification (perhaps merely promoted from the appendix) that demonstrates more nuanced aspects of superiority vs. LA independent from the specific choices of quality functions and attacker functions used in the original LA formulation (unless those are somehow thought to be essential to the approach in a manner I'm missing).

### Changes I'd recommend for Future Submissions:
_What changes do I recommend for future submissions, but would not necessarily change my score now?_

Significantly rework results presentation to better highlight in a single pass why DPA is better.

---

> ### Author Rebuttal · Authors · 2025-07-30
>
> We thank you for your inquisitive and candid review. Please find our responses below.
>
> $\newline$
>
> **Key Weaknesses** - Flaw in your results:
>
> The result from our masking experiment is correct. The non-monotonic behavior shown in LA is a result of its conflation of biases from the $A\rightarrow T$ and $T\rightarrow A$ directions.
>
> In the aforementioned experiment, we progressively mask the person to monotonically decrease the $A\rightarrow T$ bias. But, this masking also leads to a monotonic increase in the $T\rightarrow A$ biases because, as the model has less information on the gender ($A$), it relies more on task ($T$) to make its predictions.
>
>
> As $DPA_{T\rightarrow A}$ measures biases only in the $T\rightarrow A$ direction, it shows a monotonic increasing behavior. On the other hand, LA confounds the biases from both directions. I.e., An increase in $T\rightarrow A$ direction may be offset by decreases in $A\rightarrow T$ direction, resulting in non-monotonic behavior. Please read the next section -  [**Key Weaknesses** - Insufficient novelty over LA] to understand why LA conflates biases.
>
> Thus, the experiment shows us the limitations of LA and how it can give misleading results. Meanwhile, DPA, being a directional metric, can give users a clearer understanding of the bias amplification.
>
> $\newline$
>
> **Key Weaknesses** - Insufficient novelty over LA:
>
> A major improvement over LA (our novelty) is this — LA measures bias amplification (or change in bias) w.r.t varying priors. DPA (our metric) measures bias amplification with respect to a fixed prior.
>
> What does LA do? To measure dataset bias, LA uses an attacker model to predict the ground-truth attribute $A$ using the ground-truth task $T$ as input. In probabilistic terms, this is proportional to $P(A|T)$. To measure model bias, LA uses an attacker model to predict the ground-truth attribute $A$ using the task predictions $\hat{T}$ as input. This is proportional to $P(A/\hat{T})$. Bias amplification in LA (model bias - dataset bias) is proportional to $P(A|\hat{T}) - P(A|T)$. Here, the prior variable changes from $\hat{T}$ to $T$. The change in bias (or bias amplification) is not grounded w.r.t a fixed variable.
>
> What does $DPA_{T\rightarrow A}$ do? To measure dataset bias, DPA is the same as LA: dataset bias in DPA is proportional to $P(A|T)$. To measure model bias, DPA uses an attacker model to predict the ground-truth predictions $\hat{A}$ using the ground-truth task $T$ as input. In probabilistic terms, this is proportional to $P(\hat{A}|T)$. Bias amplification in DPA (model bias - dataset bias) is proportional to $P(\hat{A}|T) - P(A|T)$. In DPA, the change in bias (or bias amplification) is grounded with respect to a fixed variable (in this case, $T$).
>
> “Fundamentally, a change in bias should be measured with respect to a fixed quantity” — this idea was conveyed by Wang et al. [1] in their paper, and we use the same idea for our work. This is our main improvement over LA (LA does not follow this idea --- thus, it conflates biases in $A \rightarrow T$ and  $T \rightarrow A$). We attempted to explain this aspect of our novelty in the Appendix, Section B. If you feel this aspect of our novelty should be highlighted better in the main paper, we would be happy to do that.
>
> Note: $P(\hat{A}|T) - P(A|T)$ is a simplified expression for DPA. The precise expression for DPA is $\sum P(T_{i})P(\hat{A_{mi}}|T_{i}) - \sum P(T_{i}) P(A_{mi}|T_{i})$. This expression also ensures a fixed prior variable. For more details, please read the section: **DPA clarification**  of our response to Reviewer bRtk)
>
> $\newline$
>
> **Minor Weakness 1**
>
> Thank you for the suggestion. We will add correlation-based scores in our tables to summarize the results in a single row. This will allow readers to quickly parse and compare different metrics.
>
> **Minor Weakness 2**
>
> Presently, we do not have a "theoretical" justification for why a relative metric is superior. We have shown the practical advantages of using relative metrics over absolute ones, like robustness to different attacker models.
>
> **Minor Weakness 3**:
>
> We have a controlled experiment on the COCO dataset as described in Appendix F. Herein, we use shortcuts created using the One Pixel Shortcut[2] technique to control the hidden bias. In addition to this experiment, we have added another experiment on the CMNIST dataset (see Controlled experiment in response to Reviewer dTPG).
>
> $\newline$
>
> **Question 1**:
> We flipped $\hat{T}$ to ensure $\hat{T}$ and  $T$ have the same accuracy value. If $\hat{T}$ and $T$ do not have the same accuracy, bias amplification would be $P(\hat{T}|A) - P(T|A) + f(\text{difference in accuracy})$. Here, $f$ is some arbitrary function. The bias amplification score should only reflect the gap between the model and the dataset bias. This score should not incorporate the gap in accuracy between the model and the dataset. If the accuracy of $\hat{T}$ is $70$%, we flipped $30$% of the labels in $T$ so that $T$ comes down to the same accuracy as $\hat{T}$. This ensures that bias amplification is $P(\hat{T}|A) - P(T|A)$.
>
> $\newline$
>
> **Question 2**:
> While we are not fully familiar with multicalibration, we understand that it is a bias mitigation technique. We feel that metrics like DPA can be used to identify models that require or could benefit from multi-calibration.
>
> $\newline$
>
> **Formatting Concern**:
> Thank you for pointing this out. For practical use, a small value epsilon ~1e-10 is added to the denominator to ensure numerical stability. We will update the manuscript to reflect this.
>
> $\newline$
>
> References
>
> [1] Angelina Wang and Olga Russakovsky. Directional bias amplification. In International Conference on Machine Learning, pages 10882–10893. PMLR, 2021.
>
> [2] Shutong Wu, Sizhe Chen, Cihang Xie, Xiaolin Huang. "One-Pixel Shortcut: On the Learning Preference of Deep Neural Networks." The Eleventh International Conference on Learning Representations. 2023.

---

> > ### Comment · Reviewer_n6n6 · 2025-08-04
> > **Response**
> >
> > Thank you for your detailed response; this clarifies my concern and I will increase my score to a borderline accept accordingly.
> >
> > However, I think that you should also endeavor to significantly improve the clarity of your presentation to reflect some of these intuitions in the main body of your paper. This is a somewhat subtle point, but nevertheless one that integrally affects the validity of your results and your framing, so it warrants clear and careful communication. It is also an essential point to help the field build on your work in the future, so it is doubly important to make these distinctions clear in the main body of the paper.
> >
> > I'm also going to reduce my confidence given that I still don't feel I fully understand things here, and because I can't see the updated view of your paper with these changes (and I think improvements to the clarity are very important), and recommend to the AC that though my score is a borderline accept, it should likely not be used as the deciding vote amongst the reviews in favor of those by more confident reviewers.
> >
> > Thank you again!

---

### Official Review · Reviewer_bRtk · 2025-07-02

**Clarity:** 3
**Significance:** 3
**Originality:** 3
**Rating:** 5
**Confidence:** 3

**Summary:**

The paper proposes DAP metric to quantify the amount by which a predictive model amplifies pre-existing correlations between sensitive attributes $A$ and labels $T$. It is demonstrated experimentally that models can amplify such correlations even when trained on curated datasets where $A$ is statistically independent from $Y$. The authors argue that this is because the model can leverage spurious correlations to infer $A$ from the image. Since alternatives to DAP do not highlight such shortcuts, DAP is a worthy addition to the literature.

**Questions:**

## Clarifications Regarding the DAP
From what I understand, the bias measures $\Psi_A^D$ consists of reporting the accuracy of an attacker $f_A^T$ trained to predict $T$ from $A$. However, in Appendix B equation (13), it is stated that $\Psi_A^D \propto P(T|A)$. I think this equation is incorrect, which undermines the connection the authors were trying to make with the
$BA_{\rightarrow}$ method . Indeed, assuming $A$ and $T$ are binary, the accuracy of the optimal attacker is $f_A^T$ is $\Phi_A^D = P(A=0) \text{max}\\{P(T=1|A=0), P(T=0|A=0)\\} + P(A=1) \text{max}\\{ P(T=1|A=1), P(T=0|A=1)\\}$. This is not proportional to a conditional probability. Accordingly, Section B should be clarified or removed.

## Clarifications regarding Section 5.1
First of all, it is stated that DAP could be used for model selection. However, I think this is only true for the $DAP_{A\rightarrow T}$ which only requires that the model outputs labels $\hat{T}$. The metric $DAP_{T \rightarrow A}$ requires that the model predicts the sentitive attribute $\hat{A}$, and I cannot think of any practical application where we would want to use such predictive model. I would argue that $DAP_{T \rightarrow A}$ is more useful for identifying spurious correlations in datasets and potentially curating them.

Also, I am confused by the toy example provided in lines 333-342. In this example, we have $A=0$ for men, $A=1$ for women, $T=0$ for hired and $T=1$ for rejected. It is claimed that this dataset is biased according to DAP. But isn't DAP an amplification metric that requires a predictive model $M$ as well? What is you model in this toy example. Also DAP is directional, so which direction is considered ($A\rightarrow T$ or $T\rightarrow A$) in this toy example? Since this toy example is brought in the conclusion, it would be important to clarify why DAP claims it is biased.

## Missing Experimental Details
There are not enough details regarding the model training on Coco. Indeed, a single image can contain multiple objects of different/same classes (Figure 1 shows an image with three different objects for instance). Is the model required to localize these objects? Is it required to differentiate two different objects of the same class (e.g. two different pans)? Or simply predict there is *at least* one pan in the image?
I suspect that you treat Coco as a *multi-label classification* problem where image $I$ is labeled as $y=(y^{(1)}, y^{(2)}, \\ldots, y^{|T|})$ with $y^{(t)}=1$ if at least one object of class $t$ is present and $y^{(t)}=0$ otherwise. This would be would be important to clarify in the manuscript along with descriptions of how the architectures pretrained on Imagenet multiclass classification were adapted for Coco.

Section I (line 1012) of the appendix does not contain enough details on the Attacker network that is used to predict $A$ from $T$. For Coco, the ground-truth label $T$ fed to the attacker is actually a **set** (see Figure 1 bottom) and not a single numerical value. MLPs are not suited to take sets as inputs so I suspect that the labels were One-Hot-Encoded to $\\{0, 1\\}^{|T|}$ before being fed to the attacker. It would be important to clarify if this is the case or not. If not, how were MLPs able to process sets of objects as their inputs?

## Fitting a MLP on COMPAS
In COMPAS, the labels and sensitive attributes are binary. Thus, I do not see any benefits to fitting a MLP to predict $A$ from $T$ (or $T$ from $A$). In this case, computing conditional probabilities from the 2x2 contingency table should be sufficient. Can the authors argue why a MLP was used as an attacker on COMPAS?

**Ethical Concerns:**

["NO or VERY MINOR ethics concerns only"]

**Final Justification:**

Increased my score due to the rebuttal.

**Limitations:**

Yes

**Quality:**

3

**Strengths And Weaknesses:**

# Strengths
- The paper is well written and easy to follow.
- The experiments consisting of progressively masking individuals and trying to infer their gender $A$ based on the ground-truth $T$ (or predicted $\hat{T}$) objects is a sound way to quantify spurious correlations in the dataset/model.
- Relevant classifications models are studied (ViT newer and VGG16 older).
- The DAP metric is a significant contribution to the community because it highlights a scenario that other metrics fail to identify. Indeed, DAP can detect when a model infers gender based on spurious correlations although the dataset is curated to render $A$ and $T$ statistically independent.
- The work is original and cites revelant prior work.

# Weaknesses
- The clarity of the technical sections could be improved (more on this below in the **Questions** section)
- The experimental details are not sufficient. Notably there are missing details on the trained classifiers $M$ and the Attackers (more on this below in the **Questions** section).


## Minor Comments
The current work investigates statistical measures of fairness so there is no notion of causality involved. However, at line 112, the word *causality* is used as a synonym for directionality. This is not the correct terminology. $A\rightarrow T$ essentially means $P(T|A)$ and
$T\rightarrow A$ refers to $P(A|T)$, but using the wording *causality* along side arrow-based notation $\rightarrow$ suggests that you are doing causal inference. I would avoid writting *causality*.

---

> ### Author Rebuttal · Authors · 2025-07-30
>
> Thank you for your insightful and detailed review.
>
> We have answered to your concerns in the responses below:
>
> $\newline$
>
> **Minor Comments**:
>
> Thank you for pointing this out. We will be removing the term “causality” from our manuscript to avoid any confusion.
>
> $\newline$
>
> **DPA clarification**:
>
> You are indeed correct, we had simplified the expression to $P(T|A)$. We will modify the manuscript to reflect the expression you provided. Note that this new expression still ensures that the metric is directional. We prove this here:
>
> As a generalization of your expression, $\Psi_{A}^{D}$ can also be written as: $\Psi_{A}^{D} = \sum_{A_{i} \in A} P(A_i) P(T_{mi}|A_{i})$.
>
> Where, $T_{mi}$ is such that $P(T_{mi}|A{i}) \geq P(T_{k}|A{i}) \forall k$.
>
>
> Similarly, $\Psi_{A}^{M} = \sum_{A_{i} \in A} P(A_i) P(\hat{T_{mi}}|A_{i})$.
>
> Hence, your final expression can be written as: $\sum_{A_{i} \in A} P(A_i) P(\hat{T_{mi}}|A_{i}) -\sum_{A_{i} \in A} P(A_i) P(T_{mi}|A_{i})$. We see that bias amplification is still measured with respect to a fixed a prior variable (in this case $A_{i}$), which is how a directional metric should work.
>
> $\newline$
>
> **Section 5.1**:
>
> Discussion Section Part 1: You have raised very relevant points. We will add the following clarifications to paragraph 1 of Section 5.1: $DPA_{ A \rightarrow T}$ can be used for model selection, and $DPA_{ T \rightarrow A}$ can be used for identifying spurious correlations in datasets.
>
> Discussion Section Part 2: While addressing your comments for the second part of Section 5.1, we found an error in how we wrote this portion. We mentioned in line 340 — “Compared to $BA_{\rightarrow}$, DPA may under-report bias amplification...” This is incorrect as $BA_{\rightarrow}$ would give a “0” bias amplification if the dataset bias is 0. To convey our point correctly, we updated the second part of Section 5.1. Please read updates here:
>
> **Table 1**
>
> |       | $T=0$       | $T=1$       |
> |-------|-------------|-------------|
> | $A=0$ | $150$      | $50$      |
> | $A=1$ | $78$      | $22$      |
>
> **Table 2**
>
> |       | $\hat{T}=0$       | $\hat{T}=1$       |
> |-------|-------------|-------------|
> | $A=0$ | $185$      | $15$      |
> | $A=1$ | $91$      | $9$      |
>
>
> "Based on our experiments, we found $DPA$ to be the most reliable metric to measure bias amplification. However, there are cases where other directional metrics like $BA_{\rightarrow}$ could be more appropriate. Let us take two scenarios where we measure $A\rightarrow T$ bias amplification in $BA_{\rightarrow}$ and $DPA$.
>
> Consider a hiring dataset where $200$ men ($A = 0$) and $100$ women ($A = 1$) apply for a job. Out of these, $50$ men and $22$ women are hired ($T = 0$), while the rest are rejected ($T = 1$) — refer to Table 1. Now, assume we train a model (M) on this dataset, which gives predictions as shown in Table 2.
>
> According to DPA, the $A\rightarrow T$ bias amplification is large. This is because the counts of $T$ and $\hat{T}$ have grown farther apart for both men {initial difference = 100 (150 - 50), new difference = 170 (185 - 15)} and women {initial difference = 56 (78 - 22), new difference = 82 (91 - 9)}. According to $BA_{\rightarrow}$, $A\rightarrow T$ bias amplification is small. This is because the ratio of hired men and hired women is similar in $T$ (25% vs 22%) and $\hat{T}$ (7.5% vs 9%). Here, our goal is to ensure that men and women have a similar acceptance ratio. Since the acceptance ratio between men and women is similar in the dataset and in the model’s predictions, the small bias amplification reported by $BA_{A\rightarrow T}$ is appropriate. In scenarios where we desire equal opportunity (e.g., hiring datasets where both men and women should have equal acceptance ratios), using $BA_{\rightarrow}$ is better as it measures bias amplification by comparing task ratios between men and women.
>
> In another scenario, consider a dataset of men ($A = 0$) and women ($A = 1$), where each person is either indoors ($T = 0$) or outdoors ($T = 1$). Let us consider the same tables for the dataset (Table 1) and the model’s predictions (Table 2). Here, our goal is to ensure that we have a similar count of men and women, indoors and outdoors. Keeping our goal in mind, we should report a high $A\rightarrow T$ bias amplification as the count of $\hat{T}$ has changed significantly with respect to $T$ (for both men and women). Here, the large bias amplification reported by $DPA_{A \rightarrow T}$ is appropriate. In scenarios where we desire equal counts across $A–T$ pairs, comparing task ratios (as done by $BA_{\rightarrow}$) is not meaningful. Instead, the count-based comparison used by DPA better reflects bias amplification.  $BA_{\rightarrow}$ and $DPA$ capture different notions of bias, and the right choice of metric depends on the type of bias we seek to address."
>
> $\newline$
>
> **Experiment Details**:
>
> You are right. We used annotations from the MS-COCO multi-label benchmark. The label $y = (y_1, y_2, …. y_{|T|})$ is a binary variable. $y_{i} = 1$ indicates that at least one instance of the $i^{th}$ object is present in the image. To adapt the ImageNet pretrained models to COCO, we removed the last MLP layer from these models and replaced it with an MLP layer of size |T|. The models were then fine-tuned on the COCO dataset. We will make sure to add these points along with more details on the model training. For the attacker models, the labels are one-hot encoded. This was not shown in the figure for the sake of simplicity. We will clarify this in the text to avoid any confusion.
>
> $\newline$
>
> **MLP on COMPAS**:
>
> Yes, the 2x2 contingency table should be sufficient in the COMPAS example. For the sake of uniformity, we chose to use the MLP as the attacker function. We can compute DPA using the 2x2 contingency matrix, if you suggest that it adds value to the study.

---

> > ### Comment · Reviewer_bRtk · 2025-08-05
> > **Response to Rebuttal**
> >
> > I wish to thank the authors for addressing my concerns and answering to my questions. I will therefore increase my score.
> >
> > Regarding the experiments on COMPAS, I think results using 2x2 contingency directly should be reported (at least in appendix to show that the MLP yields the same results). This is because the optimal classifier is trivially computed on a 2x2 contingency table. Training MLPs comes with unnecessary computational costs.

---

### Author Response · Authors · 2025-08-08

We would like to thank all the reviewers for their time and effort in reviewing our work. We are delighted to see that the reviewers appreciated the novelty and impact of our proposed method, along with the diversity of our experiments.

Your reviews have helped immensely improve the clarity of the paper and gave us the opportunity to add a new controlled experiment on the CMNIST dataset to further improve robustness. We are glad we were able to address all the major concerns expressed by the reviewers.

Here is a summary of the current scores as we understand them from communications with reviewers:

* Reviewer bRtk: 5
* Reviewer n6n6: 4
* Reviewer SYXG: 4
* Reviewer dTPG: 4

---

### Decision · Program_Chairs · 2025-09-17

**Decision:**

Accept (poster)

**Comment:**

The authors tackle the problem of quantifying bias amplification in classification datasets. They proposes Directional Predictability Amplification (DPA), a predictability-based metric for directional bias amplification, which differs from prior work by working on balanced datasets and being able to detect negative bias amplification. The authors empirically demonstrates the superiority of DPA over existing metrics on three datasets.

**Strengths**

The problem is impactful and timely. The paper is well-written and easy to follow, with the formulation of the metric being clear and sound. Reviewers were generally happy with the experimental evaluations.

**Initial Weaknesses**

1. SYXG and bRtk asked for stronger evidence that DPA is minimally sensitive to attacker choice and for broader attributes/domains. SYXG also requested computational cost details.
2. n6n6 raised some concerns about novelty over LA.
3. n6n6 felt that the metric could be more theoretically principled, and SYXG asked for better theoretical characterization of potential failure modes.
4. bRtk and n6n6 raised issues about clarity in various parts of the paper.


**Rebuttal Period**

The authors provided clarifications on novelty over LA and LA non-monotonicity which satisfied Reviewer n6n6. They added a CMNIST experiment in response to Reviewer dTPG, and provided a runtime analysis for Reviewer SYXG. They also promised various clarity improvements in the revision.



**Overall Evaluation**

The paper fills a gap in the literature by providing a metric that is directional, sign-aware, and valid on balanced and unbalanced datasets. The empirical evidence is convincing. Reviewers raised valid concerns about lack of theoretical justification for some aspects (e.g. failure modes, minimal sensitivity to attacker function choice, more formal justification for dominance over LA) which have not been addressed, but I do not think this is significant enough for rejection. All reviewers eventually converged to acceptance. As such, I recommend acceptance. The authors should make the promised changes, as well as run the 2x2 contingency table experiment on COMPAS suggested by Reviewer bRtk, for the camera ready.